# Leveraging auxiliary data from arbitrary distributions to boost GWAS discovery with Flexible cFDR

**Anna Hutchinson**[1]*, **Guillermo Reales**[2,3], **Thomas Willis**[1], **Chris Wallace**[1,2,3]

**1** MRC Biostatistics Unit, University of Cambridge, Cambridge, United Kingdom, **2** Cambridge Institute of Therapeutic Immunology and Infectious Disease (CITIID), University of Cambridge, Cambridge, United Kingdom, **3** Department of Medicine, University of Cambridge, Cambridge, United Kingdom

* anna.hutchinson@mrc-bsu.cam.ac.uk

**Data Availability Statement:** All the data underlying this study are publicly available at https://doi.org/10.5281/zenodo.5554628. This resource contains the full results for Applications 1 and 2. The methods were run using the following

## Abstract

Genome-wide association studies (GWAS) have identified thousands of genetic variants that are associated with complex traits. However, a stringent significance threshold is required to identify robust genetic associations. Leveraging relevant auxiliary covariates has the potential to boost statistical power to exceed the significance threshold. Particularly, abundant pleiotropy and the non-random distribution of SNPs across various functional categories suggests that leveraging GWAS test statistics from related traits and/or functional genomic data may boost GWAS discovery. While type 1 error rate control has become standard in GWAS, control of the false discovery rate can be a more powerful approach. The conditional false discovery rate (cFDR) extends the standard FDR framework by conditioning on auxiliary data to call significant associations, but current implementations are restricted to auxiliary data satisfying specific parametric distributions, typically GWAS *p*-values for related traits. We relax these distributional assumptions, enabling an extension of the cFDR framework that supports auxiliary covariates from arbitrary continuous distributions ("Flexible cFDR"). Our method can be applied iteratively, thereby supporting multi-dimensional covariate data. Through simulations we show that Flexible cFDR increases sensitivity whilst controlling FDR after one or several iterations. We further demonstrate its practical potential through application to an asthma GWAS, leveraging various functional genomic data to find additional genetic associations for asthma, which we validate in the larger, independent, UK Biobank data resource.

## Author summary

Genome-wide association studies (GWAS) detect regions of the human genome that are associated with various traits, including complex diseases, but the power to detect these genomic regions is currently limited by sample size. The conditional false discovery rate (cFDR) provides a tool to leverage one GWAS study to improve power in another. The motivation is that if two traits have some genetic correlation, then our interpretation of a low but not significant *p*-value for the trait of interest will differ depending on whether

data: 1) Publicly available GWAS summary statistics (https://www.ebi.ac.uk/gwas/studies/GCST006862). 2) Publicly available GenoCanyon scores (http://zhaocenter.org/GenoCanyon_Downloads.html). 3) Publicly available H3K27ac ChIP-seq data from NIH Roadmap (https://egg2.wustl.edu/roadmap/data/byFileType/signal/consolidated/macs2signal/foldChange/). All other relevant data are available within the manuscript and its Supporting information files.

**Funding:** AH is funded by the Engineering and Physical Sciences Research Council (EPSRC) https://epsrc.ukri.org/ (EP/R511870/1) and GlaxoSmithKline (GSK) https://www.gsk.com/. CW is funded by the Wellcome Trust https://wellcome.ac.uk/ (WT107881), the Medical Research Council (MRC) https://mrc.ukri.org/ (MC UU 00002/4) and supported by the NIHR Cambridge BRC https://cambridgebrc.nihr.ac.uk/ (BRC-1215-20014). GR is funded by the Wellcome Trust https://wellcome.ac.uk/ (WT107881). TW is funded by the Medical Research Council (MRC) https://mrc.ukri.org/ (MC UU 00002/4). The funders had no role in study design, data collection and analysis, decision to publish, or preparation of the manuscript. The views expressed are those of the author(s) and not necessarily those of the NHS, the NIHR or the Department of Health and Social Care. For the purpose of open access, the author has applied a CC BY public copyright licence to any Author Accepted Manuscript version arising from this submission.

**Competing interests:** I have read the journal's policy and the authors of this manuscript have the following competing interests: Anna Hutchinson is a GSK-sponsored iCASE student.

that SNP shows strong or absent evidence of association with the related trait. Here, we describe an extension to the cFDR framework, called "Flexible cFDR", that controls the FDR and supports auxiliary data from arbitrary distributions, surpassing current implementations of cFDR which are restricted to leveraging GWAS $p$-values from related traits. In practice, our method can be used to iteratively leverage various types of functional genomic data with GWAS data to increase power for GWAS discovery. We describe the use of Flexible cFDR to supplement data from a GWAS of asthma with auxiliary data from functional genomic experiments. We identify associations novel to the original GWAS and validate these discoveries with reference to a larger, more highly-powered GWAS of asthma.

## Introduction

Genome-wide association studies (GWAS) identify risk loci for a phenotype by assaying single nucleotide polymorphisms (SNPs) in large participant cohorts and marginally testing for associations between each SNP and the phenotype of interest. Conducting univariate tests for each SNP in parallel presents a huge multiple testing problem for which a stringent $p$-value threshold is required to confidently call significant associations.

The statistical power to detect associations can be increased by leveraging relevant auxiliary covariates. For example, pervasive pleiotropy throughout the genome [1] suggests that leveraging GWAS test statistics for related traits may be beneficial, whilst the non-random distribution of trait-associated SNPs across various functional categories [2] suggests that incorporating functional genomic data may also be useful. In fact, the expansive range of relevant auxiliary covariates has accumulated in a wealth of covariate-informed multiple testing methods which leverage auxiliary covariates (e.g. SNP-level data) with test statistics for variables (e.g. GWAS $p$-values for SNPs) to increase statistical power. These methods have been extensively researched both in the statistical literature [3–13] and specifically in the context of GWAS [14–22] with a consistent aim of minimizing type-2 errors (or equivalently increasing statistical power) whilst controlling some appropriate type-1 error rate, such as the false discovery rate (FDR).

The conventional multiple testing correction procedures control some error measure by assuming that each hypothesis is equally likely *a priori* to be true or false. For example, the Bejamini-Hochberg (BH) procedure [23] provides nearly optimal control of the FDR under the condition that almost all null hypotheses are true [10]. The simplest extension to incorporate covariates is independent filtering [9] whereby test statistics are first censored based on the value of the covariate and a multiple testing method (for example, the BH procedure) is then applied on the remaining subset of test statistics. Alternatively, the test statistics can be grouped based on covariate values and the BH procedure can be applied separately within each group, as in stratified FDR [4]. However, these simplistic approaches do not make full use of the information contained within the covariate values and require subjective covariate thresholding which may lead to data dredging bias [6].

Insinuating a more unified approach of incorporating covariates, the conventional multiple testing correction procedures can incorporate weights for each variable, such that the raw test statistics are no longer considered exchangeable [24, 25]. However, it is non-trivial to convert covariate values to weights satisfying certain constraints required for type-1 error rate control (typically non-negative weights that average 1 [19]), and consequently the covariate values are typically still only used in an initial stratification step. For example, the grouped Benjamini-

Hochberg (GBH) procedure [5] groups test statistics based on covariate values and derives group-specific weights that are proportional to the estimated proportion of true nulls in each group, whilst the independent hypothesis weighting (IHW) method calculates optimal group-specific weights which maximise the number of discoveries, whilst controlling the FDR [6]. In the GWAS setting, the FINDOR method (which was shown to be generally less powerful, but superior in terms of false positive findings, to stratified FDR, GBH and IHW methods) was developed to leverage auxiliary data relating to how well SNPs tag functional categories that are enriched for heritability with GWAS test statistics [15, 26, 27]. SNPs are grouped based on the auxiliary data and group-specific weights, that are proportional to the ratio of the estimated proportion of alternative to null SNPs in each group [5, 28], are derived for use in a weighted Bonferroni procedure [3]. The FINDOR methodology is similar to that of the GBH procedure, but includes an additional step whereby the weights are normalised to average 1. This normalisation step is significant because Roeder et al. [19] demonstrated that using a data-dependent weighting scheme with weights that average to 1 preserves control of type-1 error with high probability if the number of weights learned is significantly less than number of hypothesis test performed [15]. Whilst grouping-based approaches are satisfactory when they capture all information provided by the covariate, as is possible in the case of categorical covariates, this assumption is restrictive in the case of continuous covariates or more complex multi-dimensional covariate spaces. Moreover, subjective thresholding and coarse binning is generally required, meaning that the entire dynamic range of the auxiliary data is often not fully exploited (for example, applying FINDOR to Biobank style data with the recommended 100 bins results in bins containing approximately 100K SNPs within which covariate values will vary).

Other notable methods in the GWAS literature include those that require subjective thresholding, such as GenoWAP [14], or those that require additional knowledge, such as the distribution of the true effect size [19–21]. GenoWAP was developed to leverage scores of SNP functionality (called "GenoCanyon scores" [29]) with GWAS test statistics and includes a thresholding step to define "functional" SNPs. Related methods in the statistical literature include those which estimate the proportion of true null hypotheses conditional on observed covariates and use these as plug-in estimates for the FDR [30], such as Boca and Leek's FDR regression [31], those utilising local false discovery rates (defined as the posterior probability that a variable is null given the observed test statistic [32]) [7, 33, 34] and those that focus power on more promising hypotheses [10, 11, 35, 36]. Specifically, the AdaPT method [10] adaptively estimates a Bayes optimal $p$-value rejection threshold, but this requires a SNP filtering stage in the GWAS context due to high computational demand [37]. A comprehensive overview of these statistical methods for covariate-informed multiple testing is provided by Ignatiadis and Huber [38].

The conditional FDR (cFDR) approach developed and applied by Andreassen and colleagues [39–43] is a natural extension to the FDR in the presence of auxiliary covariates. This intuitive approach mitigates many of the aforementioned shortcomings: it does not bin variables and thus makes full use of the dynamic range of covariate values, it does not include any subjective thresholding, and does not require the definition of a normalised weighting scheme. However, it was designed for a very specific setting, that is to increase GWAS discovery (in the "principal trait") by leveraging GWAS test statistics from a genetically related ("conditional") trait.

We were interested to see whether a more general form of cFDR could address the same covariate-informed multiple testing problems as the range of methods listed above. Here, we describe "Flexible cFDR", a new cFDR framework [44] that enjoys all of the benefits of the conventional cFDR approach but now supports continuous auxiliary data from arbitrary

distributions, thus enabling broader applicability. Specifically, our computationally efficient approach extends the usage of cFDR beyond only GWAS to the accelerating field of functional genomics and can be applied iteratively to incorporate additional layers of data. Integrating functional genomic data with GWAS test statistics is not a new concept, and is motivated by SNP enrichment studies [2, 16, 45, 46]. For example, Pickrell [16] found that loci that associated with serum high-density lipoprotein (HDL) concentrations were enriched for several functional annotations, including enhancer regions in HepG2 cells and coding exons. When using their "fgwas" model to integrate these functional annotations with GWAS test statistics, they were able to identify new loci that robustly associated with HDL concentrations. However, the fgwas approach outputs posterior probabilities rather than *p*-values, as are typical in GWAS, and currently only supports binary auxiliary data. In contrast, Flexible cFDR outputs quantities analogous to *p*-values and also supports a wide range of auxiliary data types, including (but not limited to) continuous data derived from functional genomic experiments (e.g. fold change values from ATAC-seq or ChIP-seq) and GWAS-related values (e.g. allele frequencies, sample sizes, Bayes factors or *p*-values).

We show through detailed simulations that Flexible cFDR increases sensitivity whilst controlling the FDR, and performs as well or better than the existing cFDR framework in the subset of use-cases supported by the latter. We demonstrate the utility of our method by leveraging a variety of functional genomic data with GWAS *p*-values for asthma [47] to prioritise new genetic associations, and compare our results to those from four existing methods which have previously been shown to outperform other approaches [15, 48]: Boca and Leek's FDR regression [31], a Bayesian approach, GenoWAP [14] and two grouping-based approaches, IHW [6] and FINDOR [15]. We evaluate results according to validation status in the larger, independent, UK Biobank data set [49].

## Materials and methods

### Conditional false discovery rate

We begin by restating the definition and empirical estimator of the conditional Bayesian false discovery rate (cFDR). Consider *p*-values for *m* SNPs, denoted by $p_1, \ldots, p_m$, corresponding to the null hypotheses of no association between the SNP and a principal trait (denoted by $H_0^p$). Let $p_1, \ldots, p_m$ be realisations from the random variable *P*. The Bayesian false discovery rate (FDR) is defined as the probability that the null hypothesis is true for a random SNP in a set of SNPs with $P \leq p$:

$$FDR(p) = Pr(H_0^p | P \leq p). \tag{1}$$

This Bayesian definition of a tail area FDR [50] is asymptotically equivalent [51] to the FDR introduced by Benjamini and Hochberg [23], which is the expected fraction of false discoveries amongst all discoveries.

Given additional *p*-values, $q_1, \ldots, q_m$, for the same *m* SNPs for a "conditional trait", the Bayesian FDR can be extended to the conditional Bayesian FDR (cFDR) by conditioning on both the principal and the conditional trait variables (in contrast to the standard FDR which conditions only on the principal trait variable). Assuming that $p_i$ and $q_i$ (for $i = 1, \ldots, m$) are independent and identically distributed (iid) realisations of random variables *P*, *Q* satisfying:

$$P | H_0^p \sim U(0, 1)$$

$$P \perp\!\!\!\perp Q | H_0^p \tag{2}$$

then the cFDR is defined as the probability $H_0^p$ is true at a random SNP given that the observed

$p$-values at that SNP are less than or equal to $p$ in the principal trait and $q$ in the conditional trait [39]. Using Bayes theorem,

$$cFDR(p, q) = Pr(H_0^p | P \leq p, Q \leq q)$$

$$= \frac{Pr(P \leq p | H_0^p, Q \leq q) \times Pr(H_0^p | Q \leq q)}{Pr(P \leq p | Q \leq q)}. \tag{3}$$

The cFDR framework implicitly assumes that there is a "positive stochastic monotonic relationship" between $p$ and $q$, meaning that on average SNPs with smaller $p$-values in the conditional trait are enriched for smaller $p$-values in the principal trait. This assumption is naturally satisfied in the typical use-case of cFDR that leverages $p$-values for genetically related traits.

Using Bayes theorem and standard conditional probability rules, Eq (3) can be simplified to:

$$cFDR(p, q) = \frac{Pr(P \leq p | H_0^p, Q \leq q) \times Pr(Q \leq q | H_0^p) \times Pr(H_0^p)}{Pr(P \leq p, Q \leq q)} \tag{4}$$

[52].

It is conventional in the cFDR literature to conservatively approximate $Pr(H_0^p) \approx 1$, and this is reasonable in the GWAS setting where the proportion of true signals is expected to be very low (this may be debateable as sample sizes increase, but it is still appropriate in terms of being conservative). Given the assumptions in Property (2), we can also approximate $Pr(P \leq p | H_0^p, Q \leq q) \approx p$, noting that this is an equality if $p$ is correctly calibrated. The estimated cFDR is therefore:

$$\widehat{cFDR}(p, q) = \frac{p \times Pr(Q \leq q | H_0^p)}{Pr(P \leq p, Q \leq q)}, \tag{5}$$

and existing methods use empirical cumulative distribution functions (CDFs) to estimate $Pr(Q \leq q | H_0^p)$ and $Pr(P \leq p, Q \leq q)$ [39, 44].

Having derived $\widehat{cFDR}$ values for each $p$-value-covariate pair, a simple rejection rule would be to reject $H_0^p(i)$ for any $\widehat{cFDR}(p_i, q_i) \leq \alpha$, for $0 < \alpha < 1$. However, as discussed in Liley and Wallace [44], $\widehat{cFDR}(p_i, q_i)$ is not monotonically increasing with $p_i$ and we do not wish to reject the null for some $(p_i, q_i)$ but not for some other pair $(p_j, q_j)$ with $q_i = q_j$ but $p_j < p_i$.

Andreassen et al. [39] use the decision rule:

$$\text{Reject } H_0^p \text{ if}: \exists \ p' \geq p_i : \widehat{cFDR}(p', q_i) \leq \alpha \tag{6}$$

which closely follows the BH procedure [23]. Yet unlike the BH procedure, this rejection rule does not control frequentist FDR at $\alpha$ [52]. Liley and Wallace [44] described a method to transform the cFDR estimates into "$v$-values", which are analogous to $p$-values and can be used to control FDR (e.g. in the BH procedure). However this approach is currently only suited to instances where the auxiliary data may be modelled using a mixture of centered normal distributions (for example by transforming auxiliary $p$-values to $Z$ scores; $q := -\phi^{-1}\left(\frac{q}{2}\right)$).

## Flexible cFDR

The cFDR estimator in Eq (5) holds in the more general setting where $q_1, \ldots, q_m$ are real continuous values from some arbitrary distribution that is positively stochastically monotonic in $p$. However, the current methods were designed with specific assumptions about the distribution of the auxiliary data ($p$-values from related traits and thus bounded by [0, 1]). Sparse data

regions are likely to be found more often in unbounded auxiliary data from arbitrary distributions (for example near the extreme observations) and empirical CDFs are typically inaccurate in sparse data regions because they are step functions. Moreover, the method used to control the frequentist FDR [44] assumes auxiliary data can be modelled using a mixture of centered Gaussian distributions, meaning that it is not yet applicable for auxiliary data from arbitrary distributions. We consequently describe a new, more versatile cFDR framework for data pairs consisting of $p$-values for the principal trait ($p$) and continuous covariates from more general distributions ($q$). We call our method "Flexible cFDR" and show that it is naturally suited to leveraging functional genomic data, which is not typically Gaussian.

**Flexible cFDR estimator.** To estimate both $Pr(Q \leq q|H_0^p)$ and $Pr(P \leq p, Q \leq q)$ in Eq (5) we first fit a bivariate kernel density estimate (KDE) using a normal kernel. To do this, we transform the $p$-values for the principal trait (derived from a two-tailed test, as is typical in GWAS) to absolute $Z$-scores ($Z_p$; since the sign of the associated $Z$-scores are essential arbitrary as they depend on which allele is designated "effect'). Since the absolute $Z$-scores are bounded by 0, the KDE will penalise the lack of negative data points and may underestimate the true density in regions close to 0. To avoid this boundary effect, we mirror the absolute $Z$-scores onto the negative real line together with their associated $Q$ values but only estimate the KDE on the non-negative part of the data, akin to the "reflection technique" described by Silverman [53]. We consequently model the PDF corresponding to $Z_p$, $Q$ in the usual way as

$$f(x, y) = \frac{1}{n} \sum_i \frac{1}{\sigma_p \sigma_q} \phi \left( \sqrt{ \left( \frac{x - \{-\phi^{-1}\left(\frac{p_i}{2}\right)\}}{\sigma_p} \right)^2 + \left( \frac{y - q_i}{\sigma_q} \right)^2 } \right) \tag{7}$$

where $\phi$ is the standard normal density and the values $\sigma_p$ and $\sigma_q$ are the bandwidths determined using a well-supported rule-of-thumb [54], which assumes independent samples. Consequently, we fit the KDE to a subset of independent SNPs in the data set (independent SNP sets can be readily found using a variety of software packages including LDAK [55] and PLINK [56]). We integrate over $P$ and $Q$ to estimate $Pr(P \leq p, Q \leq q)$.

Hard thresholding is used to approximate the distribution of $Q|H_0^p$ by $Q|P > 1/2$ in the earlier cFDR methods [39, 44]. Instead, in Flexible cFDR we empirically evaluate the influence of specific $p$-value quantities on the null hypothesis by utilising local false discovery rates, which estimate $Pr(H_0^p|P = p)$ [32]. We approximate $Pr(H_0^p|P = p, Q = q) \approx Pr(H_0^p|P = p)$ assuming that the majority of information about $H_0^p$ is contained in $P$ so that

$$Pr(P = p, Q = q, H_0^p) = Pr(H_0^p|P = p, Q = q) \times Pr(P = p, Q = q)$$
$$\approx Pr(H_0^p|P = p) \times Pr(P = p, Q = q) \tag{8}$$

where $Pr(P = p, Q = q)$ is estimated from our bivariate KDE and $Pr(H_0^p|P = p)$ is estimated using the local false discovery rate. In order to avoid boundary effects, we mirror the absolute $Z$-scores onto the negative real line and extract only the local false discovery rates for the non-negative part of the data, utilising the `locfdr` R package (https://cran.r-project.org/web/packages/locfdr/index.html) to do this.

We therefore have that

$$\widehat{Pr(Q = q|H_0^p)} = \frac{\widehat{Pr(Q = q, H_0^p)}}{\widehat{Pr(H_0^p)}}, \tag{9}$$

where $\widehat{Pr(Q = q, H_0^p)}$ is derived by integrating Eq (8) over P and $\widehat{Pr(H_0^p)}$ is derived by

integrating Eq (8) over P and Q. We then integrate over Q to obtain

$$\widehat{Pr(Q \le q | H_0^p)} = \frac{\widehat{Pr(Q \le q, H_0^p)}}{\widehat{Pr(H_0^p)}}. \tag{10}$$

where we use ^ to denote that these are estimates under the assumption $H_0^p \perp\!\!\!\perp Q | P$.

Our final cFDR estimator is therefore:

$$\widehat{cFDR}(p,q) = \frac{p \times \widehat{Pr(Q \le q | H_0^p)}}{\int_{-\infty}^{q} \int_{z_p}^{\infty} f(x,y) \, dx \, dy}. \tag{11}$$

where $z_p$ is the $Z$-score associated with $p$.

As in the conventional cFDR approach, our estimator implicitly assumes a positive stochastic monotonic relationship between $p$ and $q$. However, this is not guaranteed for the more general covariates that can now be leveraged with Flexible cFDR. If instead this relationship is negative (such that low $p$-values are enriched for high values of $q$), then the sign of the auxiliary data values can simply be reversed and the method can proceed as usual.

**Mapping p-value-covariate pairs to v-values.**   We describe a similar approach to that by Liley and Wallace [44] but remove the restrictive parametric assumptions placed on the auxiliary data.

Following Liley and Wallace [44], we define "L-regions" as the set of points with $\widehat{cFDR} \le \alpha$ and the "L-curve" as the rightmost border of the L-region, found through calculating $\widehat{cFDR}$ values for $p, q$ pairs defined using a two-dimensional grid of $p$ and $q$ values. For each observed $p_i, q_i$ pair we find the L-curve, which corresponds to the contour of estimated $\widehat{cFDR} = \widehat{cFDR}(p_i, q_i)$. We then define the L-region from this L-curve.

We derive $v$-values, which are essentially the probability of a newly-sampled realisation $(p, q)$ of $P, Q$ falling in the L-region under $H_0^p$. These are readily calculable by integrating the PDF of $P, Q | H_0^p$, denoted by $f_0(p,q)$, over the L-region:

$$v(p,q) = Pr((P,Q) \in L(p,q) | H_0^p) = \int_{L(p,q)} f_0(p,q) \, dp \, dq \tag{12}$$

[44]. In the original method, $f_0(p,q)$ is estimated using a mixture-Gaussian distribution, but to support auxiliary data from arbitrary distributions (where the only distributional constraint is that it is positively stochastically monotonic in $p$) we utilise the assumptions in Property (2) to write $f_0(p,q) = f_0^q(q)$ (since the PDF of $p$ conditional on $H_0^p$ is the standard uniform density). We estimate $f_0^q(q)$ as an intermediate step in the derivation of $\widehat{Pr(Q \le q | H_0^p)}$ (Eq 9).

The $v$-value, $v_i$, can be interpreted as the probability that a randomly-chosen $(p, q)$ pair has an equal or more extreme $\widehat{cFDR}$ value than $\widehat{cFDR}(p_i, q_i)$ under $H_0^p$ and is thus analogous to a $p$-value. We refer readers to Theorem 3.1 and its accompanying proof in Liley and Wallace (2021) [44] which shows that the $v$-values are uniformly distributed under the null hypothesis for $X = (p_i, q_i) \in [0, 1]^2$, and this naturally holds for Flexible cFDR where $X = (p_i, q_i) \in [0, 1] \times [q_{low}, q_{high}]$ (where $q_{low}$ and $q_{high}$ are the lower and upper limits of the KDE support respectively).

Deriving $v$-values, which are analogous to $p$-values, means that the output from Flexible cFDR can be used directly in any conventional error rate controlling procedure, such as the BH method [23]. The derivation of $v$-values also allow for iterative usage, whereby the $v$-values from the previous iteration are used as the "principal trait" $p$-values in the current iteration [44]. This allows users to incorporate additional layers of auxiliary data into the analysis at

each iteration, akin to leveraging multi-dimensional covariates, and this approach is exemplified both in our simulation analysis and in our application utilising ChIP-seq data.

**Adapting to sparse data regions.**    To ensure that the integral of the KDE approximated in our method equals 1, we define the limits of its support to be 10% wider than the range of the data. This however introduces a sparsity problem whereby the data required to fit the KDE in or near these regions is very sparse. Adaptive KDE methods that find larger value bandwidths for these sparser regions are computationally impractical for large GWAS data sets. Instead, we opt to use left-censoring whereby all $q < q_{low}$ are set equal to $q_{low}$ and the value for $q_{low}$ is found by considering the number of data points required in a grid space to reliably estimate the density (S1 Fig). Note that since our method utilises cumulative densities, the sparsity of data for extremely large $q$ is not an issue.

Occasionally, in regions where $(p, q)$ are jointly sparse, the $v$-value can appear extreme compared to the $p$-value. To avoid artifactually inflating evidence for association, we fit a spline to $log_{10}(v/p)$ against $q$ and calculate the distance between each data point and the fitted spline, mapping the small number of outlying points back to the spline and recalculating the corresponding $v$-value as required (S2 Fig).

**Flexible cFDR software.**    We have created an R package, `fcfdr`, that implements the Flexible cFDR method (https://github.com/annahutch/fcfdr). The software web-page (https://annahutch.github.io/fcfdr/) contains fully reproducible vignettes which illustrate how the Flexible cFDR method can be used to generate $v$-values from GWAS $p$-values and auxiliary data, and how these can be used directly in any error rate controlling procedure (for example using the `p.adjust` function with `method="BH"` for FDR-adjusted $p$-values).

Flexible cFDR supports a wide range of auxiliary data types and is particularly suited to leveraging functional genomic data, which is not typically Gaussian (e.g. fold change values from ChIP-seq or per-SNP scores of functionality). We include vignettes (https://annahutch.github.io/fcfdr/articles/extra-information.html; https://annahutch.github.io/fcfdr/articles/t1d_app.html) exemplifying the types of functional genomic data that can be leveraged and also describing how the LDAK method [55] can be used to generate an independent subset of SNPs for input into the software.

## Simulations

We used simulations to assess the performance of Flexible cFDR when iteratively leveraging various types of auxiliary data. We validated Flexible cFDR against the existing framework, which we call "Empirical cFDR" [44], in two cases where $q \in [0, 1]$ (as required by Empirical cFDR). We then evaluated the performance of Flexible cFDR in three novel use-cases where the auxiliary data is no longer restricted to [0, 1]. We also analysed the simulation data using Boca and Leek's FDR regression (referred to as BL) [31], which was the only other method that allowed for multiple covariates of this nature.

**Simulating GWAS results ($p$).**    We first simulated GWAS $p$-values for the arbitrary "principal trait" to be used as $p$ in our simulations. We collected haplotype data for 3781 individuals from the UK10K project (REL-2012–06-02) [57] for 80,356 SNPs with minor allele frequency (MAF) $\geq 0.05$ residing on chromosome 22. We split the haplotype data into 24 LD blocks representing approximately independent genomic regions defined by the LD detect method [58], and further stratified these so that no more than 1000 SNPs were in each stratification. For each stratification, we sampled 2, 3 or 4 causal variants with log odds ratio (OR) effect sizes simulated from the standard Gaussian prior used for case-control genetic fine-mapping studies, $N(0, 0.2^2)$ [59] (the mean number of simulated causal variants in each simulation was 54). We then used the `simGWAS` R package [60] to simulate $Z$-scores from a GWAS in each region

for 5000 cases and 5000 controls. We collated the $Z$-scores from each region and converted these to $p$-values representing the evidence of association between the SNPs and the arbitrary principal trait.

To generate an independent subset of SNPs required to fit the KDE, we converted the haplotype data to genotype data and used the `write.plink` function [56] to generate the files required for the LDAK software [55]. We generated LDAK weights for each of the SNPs and used the subset of SNPs with non-zero LDAK weights as an independent subset of SNPs (an LDAK weight of zero means that its signal is (almost) perfectly captured by neighbouring SNPs) [61]. Over the restricted interval of MAF values considered (MAF $\geq 0.05$), we found that the MAF distributions of the whole SNP set and the independent subset were largely comparable, so we did not here perform the MAF matching procedure discussed below in our analysis of asthma data.

**Simulating auxiliary data (q).** We consider five use-cases of cFDR (simulations A-E), defined by (i) the distribution of the auxiliary data $q$ (ii) the relationship between $p$ and $q$ and (iii) the relationship between different $q$ in each iteration (5 realisations of $q$ were sampled in each simulation representing multi-dimensional covariates so that cFDR could be applied iteratively) (Table 1). We denote the value of $q$ at SNP $i$ in realisation $k$ as $q_i^{(k)}$.

In simulation A, we sampled $q_i \sim Unif(0, 1)$ to represent iterating over null $p$-values (S3A Fig). In simulation B, we investigated the standard use-case of cFDR by iterating over $p$-values from "related traits" (S3B Fig). To do this, we used the `simGWAS` R package [60] to simulate $p$-values, specifying the shared causal variants such that each pair of vectors $p$, $q$ were guaranteed to share causal variants in exactly 4 of the 24 LD blocks, whilst each pair of vectors $q^{(k)}$, $q^{(j)}$ were expected to share causal variants in 4 of the 24 LD blocks.

In simulations C-E, we simulated auxiliary data representing functional genomic data sampled from arbitrary distributions and which varied based on dependence structure with the principal trait $p$-values. In simulation C, we sampled $q_i$ from a bimodal mixture normal distribution that was independent of $p_i$: $q_i \sim 0.5 \times N(-2, 0.5^2) + 0.5 \times N(3, 2^2)$ (S3C Fig). In simulations D and E we simulated continuous auxiliary data that was dependent on $p_i$ by first defining "functional SNPs" as causal variants plus any SNPs within 10,000-bp (to incorporate SNPs residing in the same arbitrary "functional mark"), and "non-functional SNPs" as the remainder. In simulation D, we then sampled $q_i$ from different mixture normal distributions for functional and non-functional SNPs:

$$q_i \sim \begin{cases} w \times N(\mu_1, 1) + (1 - w) \times N(\mu_2, 0.5^2), & \text{if SNP } i \text{ is non-functional} \\ \\ (1 - w) \times N(\mu_1, 1) + w \times N(\mu_2, 0.5^2), & \text{if SNP } i \text{ is functional} \end{cases} \tag{13}$$

where $\mu_1 \in \{2.5, 3, 4\}$, $\mu_2 \in \{-1.5, -2, -3\}$, $w \in \{0.6, 0.7, 0.8, 0.9, 0.95\}$ vary across iterations.

**Table 1. Summary of simulation analysis.**

| Simulation | Distribution of $q$ | Relationship between $p_i$ and $q_i^{(k)}$ | Average pairwise correlation[a] between $q^{(k)}$ and $q^{(l)}$ |
|---|---|---|---|
| A | $p$-values (all null) | Independent | 0 |
| B | $p$-values (related trait) | Shared causal variants | 0.04 |
| C | Functional | Independent | 0 |
| D | Functional | Dependent | 0.08 |
| E | Functional | Dependent | 0.19 |

[a] Pairwise correlation values are the Pearson correlation coefficients.

Since we anticipate our method being used to leverage functional genomic data iteratively, we also evaluated the impact of repeatedly iterating over auxiliary data that captures the same functional mark. To do this, in simulation E we iterated over realisations of $q$ that are sampled from the same distribution,

$$
q_i \sim \begin{cases} N(3, 2^2), & \text{if SNP } i \text{ is non-functional} \\ \\ N(-2, 0.5^2), & \text{if SNP } i \text{ is functional.} \end{cases} \tag{14}
$$

**Running empirical cFDR, Flexible cFDR and BL.**   Following the vignette for the Empirical cFDR software (https://github.com/jamesliley/cfdr/blob/master/vignettes/cfdr_vignette.Rmd), we first used the `vl` function to generate L-curves. As recommended in the documentation and to ensure that the rejection rules were not being applied to the same data from which they were determined, we used the leave-one-out-procedure whereby L-curves were fit separately for data points in each LD block using data points from the other LD blocks. To ensure that the cFDR curves were strictly decreasing (preventing a complication whereby all $v$-values corresponding to the smallest $p$-values were given the same value), we reduced the value of the `gx` parameter to the minimum $p$-value in the LD block. We then estimated the distribution of $P, Q|H_0^p$ using the `fit.2g` function and integrated its density over the computed L-regions using the `il` function, specifying a mixture Gaussian distribution for the $Z$-scores.

Flexible cFDR was implemented using the `flexible_cfdr` function in the `fcfdr` R package with default parameter values. Both cFDR methods were applied iteratively 5 times in each simulation to represent leveraging multi-dimensional covariates.

For BL, we used the `lm_qvalue` function in the `swfdr` Bioconductor R package (version 1.16.0 https://github.com/leekgroup/swfdr) to derive adjusted $p$-values. The covariate matrix that we used consisted of five columns for $q^{(1)}, q^{(2)}, q^{(3)}, q^{(4)}, q^{(5)}$.

**Evaluating sensitivity, specificity and FDR control.**   To quantify the results from our simulations, we used the BH procedure to derive FDR-adjusted $v$-values from empirical and Flexible cFDR (which we call "FDR values" for conciseness). For BL, we used the adjusted $p$-values as the quantity of interest. We then calculated proxies for the sensitivity (true positive rate) and the specificity (true negative rate) at an FDR threshold of $\alpha = 5 \times 10^{-6}$, which roughly corresponds to the genome-wide significance $p$-value threshold of $5 \times 10^{-8}$ (S4 Fig). We defined a subset of "truly associated SNPs" as any SNPs with $r^2 \geq 0.8$ with any of the causal variants. Similarly, we defined a subset of "truly not-associated SNPs" as any SNPs with $r^2 \leq 0.01$ with all of the causal variants. (Note that there are 3 non-overlapping sets of SNPs: "truly associated", "truly not-associated" and neither of these). We calculated the sensitivity proxy as the proportion of truly associated SNPs that were called significant and the specificity proxy as the proportion of truly not-associated SNPs that were called not significant. To examine whether our results were robust to using different $r^2$ values to define truly associated and truly not-associated SNPs, we also evaluated our sensitivity and specificity proxies using larger $r^2$ values.

To assess whether the FDR was controlled within a manageable number of simulations, we raised $\alpha$ to 0.05 and calculated the proportion of SNPs called FDR significant which were truly not-associated (that is, $r^2 \leq 0.01$ with all of the causal variants).

## Application to asthma

We demonstrate the utility of our method by leveraging a variety of functional genomic data with GWAS $p$-values for asthma [47]. Specifically, we describe two applications: (1) leveraging

GenoCanyon scores with asthma GWAS $p$-values and (2) leveraging ChIP-seq data in relevant cell types with asthma GWAS $p$-values. We compare the performance of Flexible cFDR to that of four existing methods in the applications which support their usage.

**Asthma GWAS data.** Asthma GWAS summary statistics for 2,001,256 SNPs were downloaded from the NHGRI-EBI GWAS Catalog [62] for study accession GCST006862 [47] on 10/10/2019. We used the $p$-values generated from a meta-analysis of several GWASs for individuals of European ancestry under a random effects model, totalling 19,954 asthma cases and 107,715 controls. The genomic inflation factor for this study was $\lambda = 1.055$, implying minimal inflation of test statistics. The UCSC liftOver utility [63] was used to convert GRCh38/hg38 into GRCh37/hg19 coordinates, and those that could not be accurately converted were removed. All co-ordinates reported are for GRCh37/hg19. We call this GWAS data the "discovery GWAS data set".

We analysed these data with methods that leverage auxiliary data as described below, and evaluated results using data from a larger asthma GWAS performed by the Neale Lab (self-reported asthma: 20002_1111) for 41,934 asthma cases and 319,207 controls from UK Biobank [49] (URL: https://www.dropbox.com/s/kp9bollwekaco0s/20002_1111.gwas.imputed_v3. both_sexes.tsv.bgz?dl=0 downloaded on 10/05/2020). Specifically, if a SNP was claimed to be significant in the discovery GWAS data set or after applying cFDR (or a comparator method), and it was also significant in the Neale Lab UK Biobank validation data, then we say that it is validated. We used the BH procedure to derive FDR-adjusted $p$-values (which we call "FDR values" for conciseness) and defined significant SNPs as those with $FDR \leq 0.000148249$, which corresponds to the genome-wide significance $p$-value threshold of $p \leq 5 \times 10^{-8}$ (0.000148249 is the maximum FDR value amongst SNPs with raw $p$-values $\leq 5 \times 10^{-8}$ in the discovery GWAS data set). We restricted analysis to the 1,968,651 SNPs that were present in both the discovery and the validation GWAS data sets.

We down-sampled the independent subset of SNPs to match the MAF distribution in this subset to that in the whole set of SNPs. This accounts for the confounding of LDAK weights and GWAS $p$-values by MAF: less common SNPs ($MAF < 0.05$) are over-represented among the independent subset and have, on average, larger $p$-values. Matching in this way prevents a bias of the KDE fit towards the behaviour of rarer SNPs. We used MAFs estimated from the CEU sub population of the 1000 Genomes resource [64], and for the 639 SNPs with missing MAF values we used values randomly sampled from the empirical MAF distribution derived from the other SNPs. This reduced the independent subset of SNPs for fitting the KDE from 509,716 to 247,879 SNPs.

To identify independent hits we used the LD clumping algorithm in PLINK 1.9 [56], using a 5 Mb window and an $r^2$ threshold of 0.01 [15]. We used haplotype data from the 503 individuals of European ancestry from 1000 Genomes project Phase 3 [64] as a reference panel to calculate LD between SNPs. The SNP with the smallest $p$-value in the discovery GWAS data set in each LD clump was called the "index SNP".

**Application 1: Leveraging GenoCanyon scores.** Tools have now been developed that integrate various genomic and epigenomic annotation data to quantify the pathogenicity, functionality and/or deleteriousness of both coding and non-coding GWAS variants [29, 65–68]. For example, GenoCanyon scores aim to infer the functional potential of each position in the human genome [29]. They are derived from the union of 22 computational and experimental annotations (broadly falling into conservation measure, open chromatin, histone modification and TFBS categories) in different cell types. We downloaded GenoCanyon scores from http://zhaocenter.org/GenoCanyon_Downloads.html for each of the 1,968,651 SNPs that GWAS $p$-values were available for, noting a bimodal distribution for the scores (S5A Fig).

**Application 2: Leveraging ChIP-seq data.**   The histone modification H3K27ac is associated with active enhancers [69] and so SNPs residing in genomic regions with high H3K27ac counts in trait-relevant cell types may be more likely to be associated with the trait of interest [70].

We downloaded consolidated fold-enrichment ratios of H3K27ac ChIP-seq counts relative to expected background counts from NIH Roadmap [71] (https://egg2.wustl.edu/roadmap/data/byFileType/signal/consolidated/macs2signal/foldChange/) in primary tissues and cells relevant for asthma: immune cells and lung tissue. We mapped each SNP in our GWAS data set to its corresponding genomic region and recorded the H3K27ac fold change values for each SNP in each cell type using bedtools intersect [72]. For SNPs on the boundary of a genomic region (and therefore mapping to two regions) we randomly selected one of the regions.

The raw H3K27ac fold change data had very long tails and so we transformed the values: $q := log(q + 1)$. We observed that the data for the different cell types roughly fell into two clusters (S6 Fig): lymphoid cells (consisting of $CD19^+$, $CD8^+$ memory, $CD4^+$ $CD25^-$ $CD45RA^+$ naive, $CD4^+$ $CD25^-$ $CD45RO^+$ memory, $CD4^+$ $CD25^+$ $CD127^-$ Treg, $CD4^+$ CD25int $CD127^+$ Tmem, CD8 naive, CD4 memory, CD4 naive and $CD4^+$ $CD25^-$ Th cells) clustered with CD56 cells whilst lung tissue clustered with monocytes ($CD14^+$ cells). We therefore averaged the transformed H3K27ac fold change values in lymphoid and CD56 cell types to derive $q1$, and the transformed H3K27ac fold change values in lung tissue and monocytes to derive $q2$, adding a small amount of noise $[N(0, 0.1^2)]$ to the latter to smooth out the discrete valued counts (S7 Fig).

## Analysis methods

**Flexible cFDR.**   We used the `flexible_cfdr` function in the `fcfdr` R package to generate $v$-values derived by leveraging the auxiliary data described above with asthma GWAS $p$-values. We defined our independent SNP set as the set of 509,716 SNPs given a non-zero LDAK weight [61] for the `indep_index` parameter and we used the optional `maf` parameter in the software to supply the MAF values required for our MAF matching procedure. We used the BH procedure to derive FDR values and used these as the output of interest.

**GenoWAP.**   GenoWAP is a Bayesian method that leverages GenoCanyon scores with GWAS $p$-values to output posterior scores of disease-specific functionality for each SNP [14]. The GenoWAP software requires a `threshold` parameter defining functional SNPs according to their GenoCanyon score. For this, we used the default recommended value of 0.1, which corresponded to 40% of the SNPs in our data set being "functional". We used the `GenoWAP.py` python script to obtain posterior scores for each SNP, and used these as the output of interest. GenoWAP could only be used for application 1 because it only supports auxiliary data that is GenoCanyon scores.

**IHW.**   Independent hypothesis weighting (IHW) [6] is a statistical method for covariate-informed multiple testing whereby variables are divided into groups and optimal group specific weights are derived (which maximise the number of discoveries whilst controlling the FDR) for use in a weighted BH procedure. We used the `IHW` Bioconductor R package (version 1.18) with default parameters, specifying the level of FDR control `alpha = 0.000148249`, and used the adjusted $p$-values as the output of interest. IHW could only be used for application 1 because it is not clear how to use this method to leverage multi-dimensional covariate vectors.

**Boca and Leek's FDR regression.**   BL estimates the proportion of null hypotheses conditional on observed covariates and uses these as plug-in estimates for the FDR [31]. We used the `swfdr` Bioconductor R package (version 1.16.0) to derive adjusted $p$-values [73] and used

these as the output of interest. For application 1, the `lm_qvalue` function was used with a covariate matrix consisting of a single column of GenoCanyon scores for each SNP. For application 2, the `lm_qvalue` function was used with a covariate matrix consisting of two columns for $q1$ and $q2$.

**FINDOR.** FINDOR is a *p*-value re-weighting method which leverages a wider range of non-cell-type-specific functional annotations. FINDOR uses the baseline-LD model from Gazal et al. [27] for prediction, and so we were unable to directly compare the methods when leveraging the same GenoCanyon or ChIP-seq auxiliary data. Instead, as recommended we used FINDOR to leverage the 96 annotations from the latest version of the baseline-LD model (version 2.2) with asthma GWAS *p*-values. Briefly, this auxiliary data contains the 75 annotations from Gazal et al. [27] (including functional regions, histone marks, MAF bins and LD-related annotations) plus extra annotations including synonymous/ non-synonymous, conserved annotations, 2 flanking bivalent TSS/ enhancer annotations from NIH Roadmap [71], promoter/ enhancer annotations [74], promoter/ enhancer sequence age annotations [75] and 11 new annotations from Hujoel et al. [76] (5 new binary annotations and corresponding flanking annotations and 1 continuous count annotation). We matched SNPs to their annotations using rsID and GRCh37/hg19 coordinates.

To run FINDOR, stratified LD score regression (S-LDSC) must first be implemented to obtain annotation effect size estimates, $\hat{\tau}_C$. To run S-LDSC, we downloaded (i) partitioned LD scores from the baseline-LD model v2.2 [27], (ii) regression weight LD scores and (iii) allele frequencies for available variants in the 1000 Genomes Phase 3 data set. We then used the `munge_sumstats.py` python script in the `ldsc` package to convert the asthma GWAS summary statistics to the correct format for use in the `ldsc` software. We restrict analysis to HapMap3 SNPs using the `-merge-alleles` flag, as recommended in the LDSC and FINDOR documentation.

We ran S-LDSC with the `-print-coefficients` flag to generate the `.result` file containing the annotation effect size estimates required for FINDOR. Specifically, pre-computed regression weight LD scores were read in for 1,187,349 variants, for which 1,034,758 remained after merging with reference panel SNP LD scores, for which 1,032,395 SNPs remained after merging with regression SNP LD scores.

To run FINDOR, partitioned LD scores must also be supplied for the SNPs in the data set. To do this, we downloaded the 1000 Genomes EUR Phase 3 PLINK files and annotation data and followed the 'LD Score Estimation Tutorial' on the LDSC GitHub page. Partitioned LD scores could be generated for the 1,976,360 (out of 2,001,256) SNPs in the asthma data set that were also present in the 1000 Genomes Phase 3 data set.

We then generated a file for the asthma GWAS data [47], including columns for sample sizes, SNP IDs and *Z*-scores. We used this file, along with the computed partitioned LD scores and the `.result` file from S-LDSC to obtain re-weighted *p*-values for the 1,968,651 SNPs using FINDOR. We used the BH procedure to convert these to FDR values and used these as our output of interest.

## Results

### Simulations show Flexible cFDR controls FDR and increases sensitivity where appropriate

We expect that leveraging irrelevant data should not change our conclusions about a study. Simulations A and C showed that the sensitivity and specificity remained stable across iterations and that the FDR was controlled at a pre-defined level when leveraging independent auxiliary data with Flexible cFDR (Fig 1A and 1C). In contrast, when leveraging relevant data, we

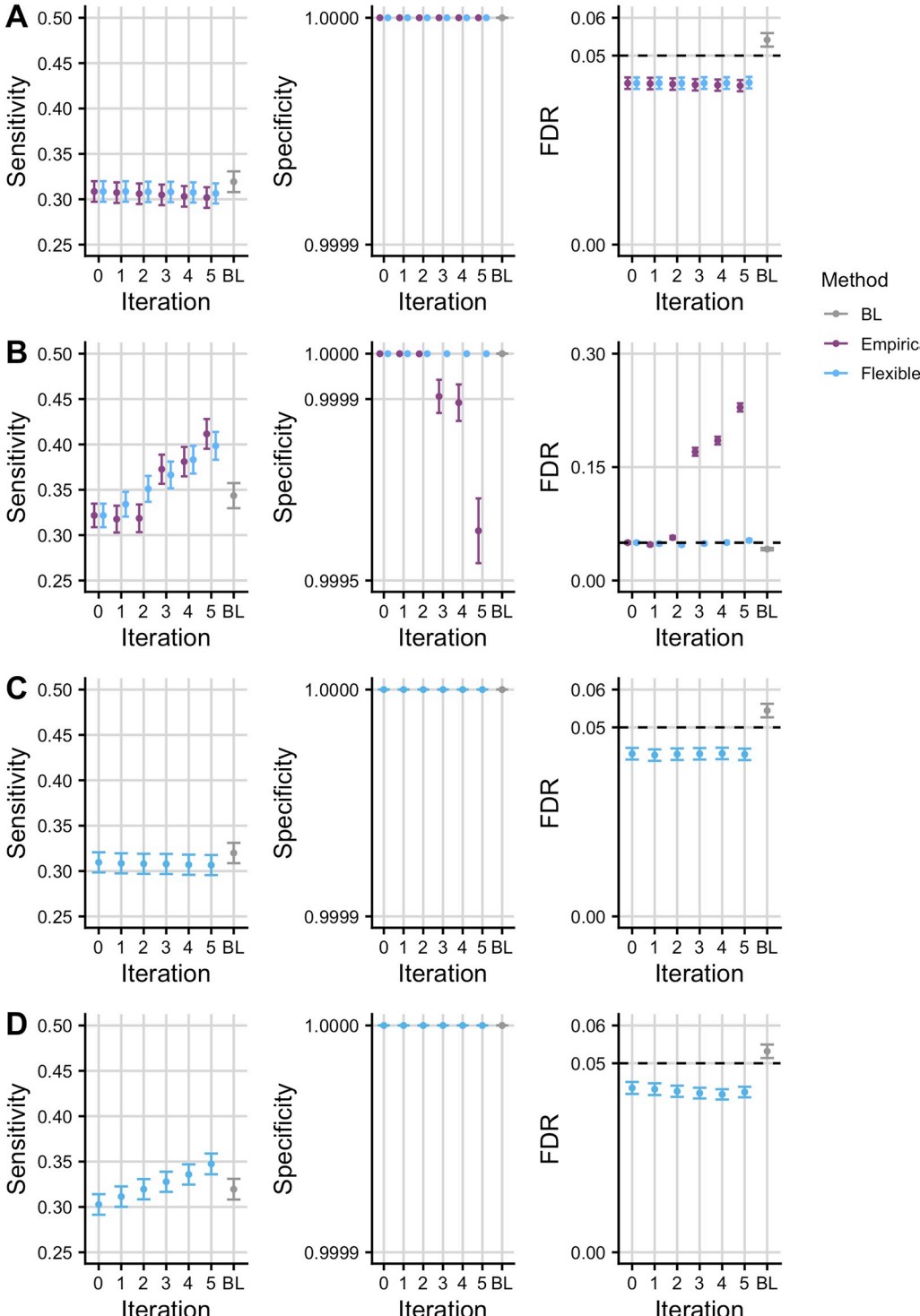

**Fig 1. Simulation results.** Mean +/- standard error for the sensitivity, specificity and FDR of FDR values from empirical and Flexible cFDR when iterating over independent (A; "simulation A") and dependent (B; "simulation B") auxiliary data that is bounded by [0, 1]. Panels C and D show the results from Flexible cFDR when iterating over independent (C; "simulation C") and dependent (D; "simulation D") auxiliary data simulated from bimodal mixture normal distributions. BL refers to results when using Boca and Leek's FDR regression to leverage the 5-dimensional covariate data. Iteration 0 corresponds to the original FDR values. Our sensitivity proxy is calculated as the proportion of SNPs with $r^2 \geq 0.8$ with a causal variant ("truly associated"), that were detected with a FDR value less than $5 \times 10^{-6}$. Our specificity proxy is calculated as the proportion of SNPs with $r^2 \leq 0.01$ with all the causal variants ("truly not-associated"), that were not

detected with a FDR value less than $5 \times 10^{-6}$. Our FDR proxy is calculated as the proportion of SNPs that were detected with a FDR value less than 0.05, that had $r^2 \leq 0.01$ with all the causal variants ("truly not-associated") (we raised $\alpha$ to 0.05 in order to assess FDR control within a manageable number of simulations). Results were averaged across 100 simulations.

hope that the sensitivity improves whilst the specificity remains high. This is what we observed for Flexible cFDR in simulations B and D (Fig 1B and 1D). The increase in sensitivity related to how informative the auxiliary data was, whereby the sensitivity generally increased more in simulation B than simulation D, where the average Pearson correlation coefficient between $p$ and $q^{(k)}$ was $r = 0.07$ and $r = 0.04$ respectively. These findings were robust to the $r^2$ threshold used to define our sensitivity (S8 Fig) and specificity (S9 Fig) proxies.

For simulations A and B, we could compare Flexible cFDR performance to that of the current method, Empirical cFDR, since $q$ could be transformed to a mixture Gaussian [44]. Performance was similar for simulation A, whilst for simulation B, the sensitivity of the two methods was comparable but Empirical cFDR exhibited a greater decrease in the specificity and failed to control the FDR in later iterations (Fig 1B). This contrasts with earlier results for Empirical cFDR, which showed good control of FDR [44], and reflects the structure of our simulations which assume dependence between different realisations of $q$.

A leave-one-out procedure is required for the Empirical cFDR method, as it utilises empirical CDFs and including an observation when estimating its own L-curve causes the curve to deviate around the observed point [44]. Flexible cFDR does not require a leave-one-out procedure as KDEs are used instead of empirical CDFs. Additionally, Flexible cFDR is quicker to run than Empirical cFDR, taking approximately 3 minutes compared to Empirical cFDR which takes approximately 6 minutes to complete a single iteration on 80,356 SNPs (using one core of an Intel Xeon E5–2670 processor running at 2.6GHz). Together, these findings indicate that Flexible cFDR performs no worse, and generally better, than Empirical cFDR in use-cases where both methods are supported.

We also benchmarked the performance of Flexible cFDR against that of BL [31], which was the only other method that allows for multiple covariates of this nature. We found that BL failed to control the FDR when leveraging independent covariate data, which may be due to the correlations between SNPs (Fig 1A and 1C). Indeed, Boca and Leek [32] found that control of the FDR by BL was worse with increasing correlation, but we note that correlation between SNPs is ubiquitous in GWAS data. When leveraging dependent covariate data, BL was consistently less powerful than Flexible cFDR (Fig 1B and 1D) and it failed to control the FDR in simulations leveraging dependent covariates from arbitrary distributions (Fig 1D), which represent the general use-case of the method [31].

We anticipate that Flexible cFDR will typically be used to leverage functional genomic data iteratively and it is helpful that specificity remains high and FDR is controlled in simulation D. It is obvious that repeated conditioning on the *same* data should produce erroneous results, with SNPs with a modest $p$ but extreme $q$ incorrectly attaining greater significance with each iteration. For strict validity, we require $q_i^{(k)} \perp\!\!\!\perp q_i^{(l)} | H_0^p$ as the $v$-value from iteration $k$ will contain some information about $q_i^{(k)}$, and the cFDR assumes $v_i \perp\!\!\!\perp q_i^{(k+1)} | H_0^p$ at the next iteration. However, even when $q_i^{(k)} \not\perp\!\!\!\perp q_i^{(l)} | H_0^p$, we expect the dependence between $v$ and $q$ to be quite weak, hence the acceptable FDR control in simulations B and D above.

Given the wealth of functional marks available for similar tissues and cell types (for example subsets of peripheral immune cells), we wanted to assess robustness of our procedure to more extreme dependence by repeatedly iterating over auxiliary data that is capturing the same functional mark. In simulation E, the sensitivity increased with each iteration at the expense of a drop in the specificity and loss of FDR control in later iterations (Fig 2). The drop in specificity

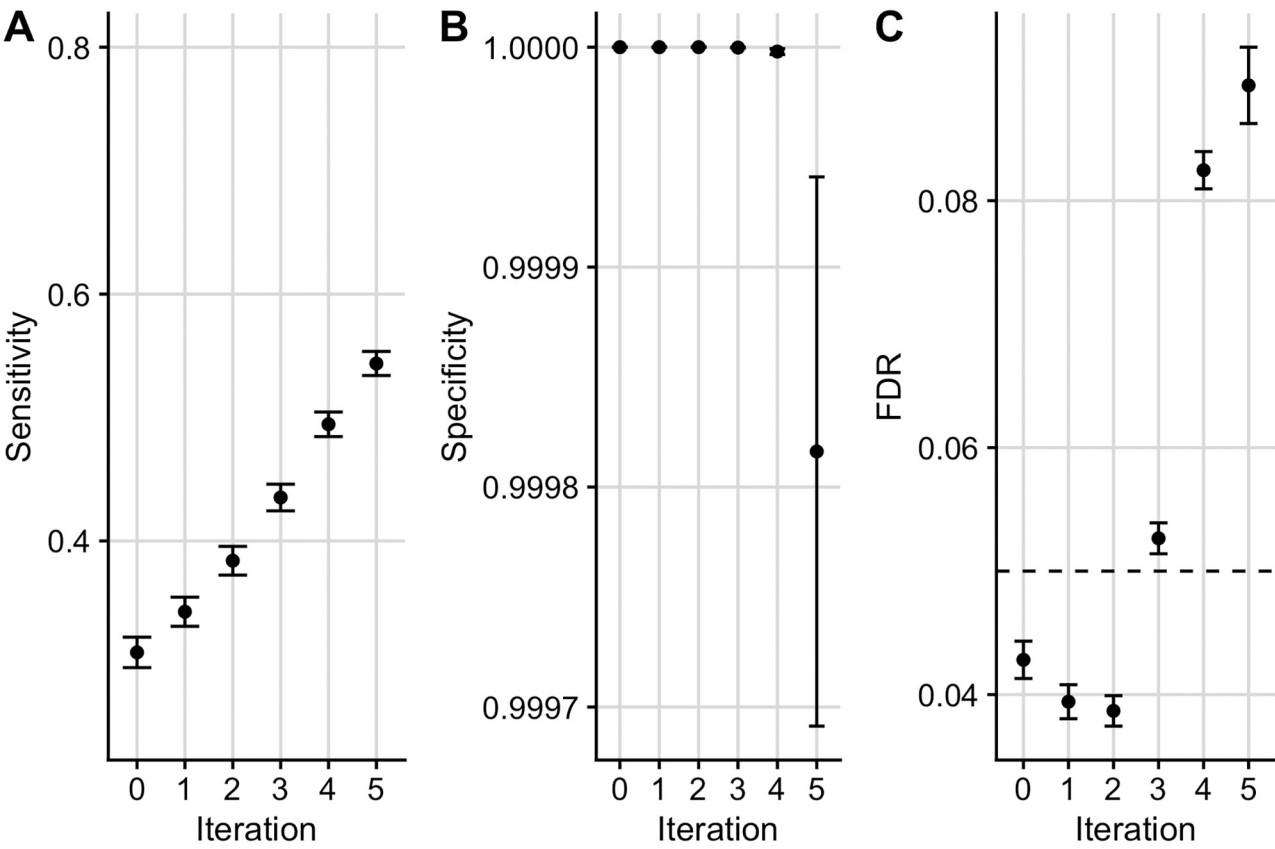

**Fig 2. Simulation results for scenario E.** Mean +/- standard error for the sensitivity (A), specificity (B) and FDR (C) of FDR values from Flexible cFDR when iterating over auxiliary data sampled from the same distribution ("simulation E"). Iteration 0 corresponds to the original FDR values. Our sensitivity proxy is calculated as the proportion of SNPs with $r^2 \geq 0.8$ with a causal variant ("truly associated"), that were detected with a FDR value less than $5 \times 10^{-6}$. Our specificity proxy is calculated as the proportion of SNPs with $r^2 \leq 0.01$ with all the causal variants ("truly not-associated"), that were not detected with a FDR value less than $5 \times 10^{-6}$. Our FDR proxy is calculated as the proportion of SNPs that were detected with a FDR value less than 0.05, that had $r^2 \leq 0.01$ with all the causal variants ("truly not-associated") (we raised $\alpha$ to 0.05 in order to assess FDR control within a manageable number of simulations). Results were averaged across 100 simulations.

and loss of FDR control is exacerbated when iterating over *exactly* the same auxiliary data in each iteration (S10 Fig), as expected. We therefore recommend that care should be taken not to repeatedly iterate over functional data that is capturing the same genomic feature, and in a real data example that follows, we average over cell types which show correlated values for functional data.

## Analysis of asthma GWAS leveraging GenoCanyon scores

Overall, 655 SNPs were FDR significant ($FDR \leq 0.000148249$) in the original asthma GWAS [47]. We used Flexible cFDR to leverage GenoCanyon scores measuring SNP functionality with asthma GWAS *p*-values and have made all of the results publicly available at https://doi.org/10.5281/zenodo.5554628. SNPs with high GenoCanyon scores were enriched for smaller asthma *p*-values (S11 Fig) and accordingly FDR values from Flexible cFDR for SNPs with high GenoCanyon scores (and therefore more likely to be functional) were lower than their corresponding original FDR values, whilst those for SNPs with low GenoCanyon scores (and therefore less likely to be functional) were higher than their corresponding original FDR values (S12 Fig). (Due to the positive stochastic monotonicity requirement for cFDR, the Flexible cFDR

software reversed the sign of the GenoCanyon scores for its internal calculations). Specifically, Flexible cFDR identified 12 newly FDR significant SNPs (rs4705950, rs6903823, rs9262141, rs1264349, rs2106074, rs3130932, rs9268831, rs3129719, rs1871665, rs16924428, rs1663687 and rs12900122) which had high GenoCanyon scores (mean GenoCanyon score = 0.77) and 3 SNPs were no longer FDR significant which had low GenoCanyon scores (mean GenoCanyon score = 0.01). At the locus level no newly significant, or newly not-significant, loci were identified.

We compared the results from Flexible cFDR to those from IHW [6], BL [31] and Geno-WAP [14] when leveraging the exact same auxiliary data and have made all of the results publicly available at https://doi.org/10.5281/zenodo.5554628. IHW groups SNPs based on their covariate values and derives optimal group-specific weights for use in a weighted BH procedure. Interestingly, all SNPs were allocated a weight of 1 in this instance, meaning that IHW reduced to the conventional BH procedure (and so the adjusted $p$-values from IHW were identical to the original FDR values from the discovery GWAS data set [47]).

In BL, logistic regression is used to estimate how the distribution of input $p$-values depends on the GenoCanyon scores to estimate the probability that the null hypothesis of no association is true for each SNP. These probabilities ranged from 0.957 for the SNP with the largest GenoCanyon score to 0.993 for the SNP with the smallest GenoCanyon score (S6B Fig). The consequence of the narrow range of these values is that the adjusted values from BL were very similar to the original FDR values (S13 Fig). Specifically, BL only identified 3 newly FDR significant SNPs, and these were all also identified by Flexible cFDR. One of these had a very high GenoCanyon score (rs1871665 with score = 0.999) whilst the other two had medium (rs16924428 with score = 0.532) or low (rs9268831 with score = 0.224) scores. No SNPs were identified as no longer FDR significant after applying BL and at the locus level, no newly significant, or newly not-significant, loci were identified.

Since GenoWAP outputs posterior probabilities rather than $p$-values, we compared the performance of the methods with GenoWAP based on the rankings of SNPs using the UK Biobank data resource. Firstly, at the SNP-level, for each of the 5152 SNPs that passed FDR significance in the UK Biobank data, we compared the rank of the FDR value in the discovery data set with (1) the rank of the FDR value after applying Flexible cFDR, (2) the rank of the FDR value from BL and (3) the rank of the (negative) posterior score from GenoWAP. (IHW was not included in this comparison because the output from IHW was just the original FDR values). We found that the percentage of FDR significant SNPs in the UK Biobank data which had an improved rank after applying each of the methods was similar (Table 2) and that 68.5% of the SNPs that improved ranks in at least one of the methods improved rank in all of the methods. Similarly, the percentage of the 1,963,499 SNPs that were not FDR significant in UK Biobank which had a decreased rank after applying each of the methods was similar (Table 2) and 49.6% of the SNPs that decreased rank in at least one of the methods decreased rank in all of the methods.

**Table 2. Summary of SNP-level results when leveraging GenoCanyon scores with asthma GWAS $p$-values.**
Table lists the percentage of the 5152 FDR significant SNPs in UK Biobank which improved rank ("UK Biobank significant which improved rank") and the percentage of the 1,963,499 SNPs that were not FDR significant in UK Biobank which decreased rank ("UK Biobank not-significant which decreased rank") after applying Flexible cFDR, BL or GenoWAP.

|  | Flexible cFDR | BL | GenoWAP |
|---|---|---|---|
| UK Biobank significant which improved rank | 60.3% | 52.4% | 61.5% |
| UK Biobank not-significant which decreased rank | 46.8% | 58.0% | 44.8% |

**Table 3. Summary of locus-level results when leveraging GenoCanyon scores with asthma GWAS *p*-values.**
Table lists the percentage of the 114 FDR significant index SNPs in UK Biobank which improved rank ("UK Biobank significant which improved rank") and the percentage of the 301 index SNPs that were not FDR significant in UK Biobank which decreased rank ("UK Biobank not-significant which decreased rank") after applying Flexible cFDR, BL or GenoWAP.

|  | Flexible cFDR | BL | GenoWAP |
|---|---|---|---|
| UK Biobank significant which improved rank | 42.1% | 28.9% | 55.3% |
| UK Biobank not-significant which decreased rank | 40.5% | 40.9% | 34.6% |

Secondly, we focused on the 114 loci that passed FDR significance in the UK Biobank data set. Within each of the 114 loci, we identified the SNP with the lowest *p*-value and called this the "index SNP". For each index SNP, we compared the rank of the FDR values in the discovery GWAS data set with (1) the rank of the FDR value after applying Flexible cFDR, (2) the rank of the FDR value from BL and (3) the rank of the (negative) posterior score from GenoWAP. We found that the percentage of UK Biobank significant SNPs (including index SNPs) that improved rank after applying Flexible cFDR was greater than that for BL (Tables 2 and 3), which matched results from our simulation analysis which showed that BL was generally less sensitive than Flexible cFDR. Similarly, the percentage of the 301 loci that were not FDR significant in UK Biobank which had a decreased rank after applying each of the methods was similar (Table 3) and 51.4% of the lead variants that decreased rank in at least one of the methods decreased rank in all of the methods.

In all, the results were similar for Flexible cFDR, BL and GenoWAP when leveraging GenoCanyon scores of SNP functionality with asthma GWAS *p*-values, but rather unexciting as no newly significant loci were identified. We suggest that this is due to the one-dimensional non-trait-specific auxiliary data that is being leveraged, which is unlikely to capture enough disease relevant information to substantially alter conclusions from a study. This is supported by our intermediary results, where the optimal weights derived in IHW were all equal to 1 and the estimated proportions of true null hypotheses conditional on the GenoCanyon scores in BL were almost negligible.

## Analysis of asthma GWAS leveraging ChIP-seq data uncovers new genetic associations

In agreement with reports that GWAS SNPs are enriched in active chromatin [77], we observed that H3K27ac fold change values in asthma relevant cell types were negatively correlated with asthma GWAS *p*-values (S14 Fig) such that SNPs with high fold change values were enriched for smaller *p*-values (S15 Fig). (Due to the positive stochastic monotonicity requirement for cFDR, the Flexible cFDR software reversed the sign of the fold change values for its internal calculations). Accordingly, FDR values from Flexible cFDR for SNPs with high H3K27ac fold-change counts in asthma relevant cell types were lower than their corresponding original FDR values, whilst those for SNPs with low H3K27ac fold-change counts in asthma relevant cell types were higher than their corresponding original FDR values (Fig 3A, 3B and 3C).

The 655 SNPs that were FDR significant ($FDR \leq 0.000148249$) in the original asthma GWAS [47] have strong replication *p*-values in the UK Biobank data set used for validation (Fig 3D; Iteration 0). By leveraging H3K27ac data, Flexible cFDR identified weaker signals that were not significant in the original data but have reassuringly small *p*-values in the UK Biobank data (median *p*-value in UK Biobank data for these SNPs is $4.65 \times 10^{-21}$; Fig 3D). Specifically, Flexible cFDR identified 51 newly significant SNPs when leveraging average H3K27ac

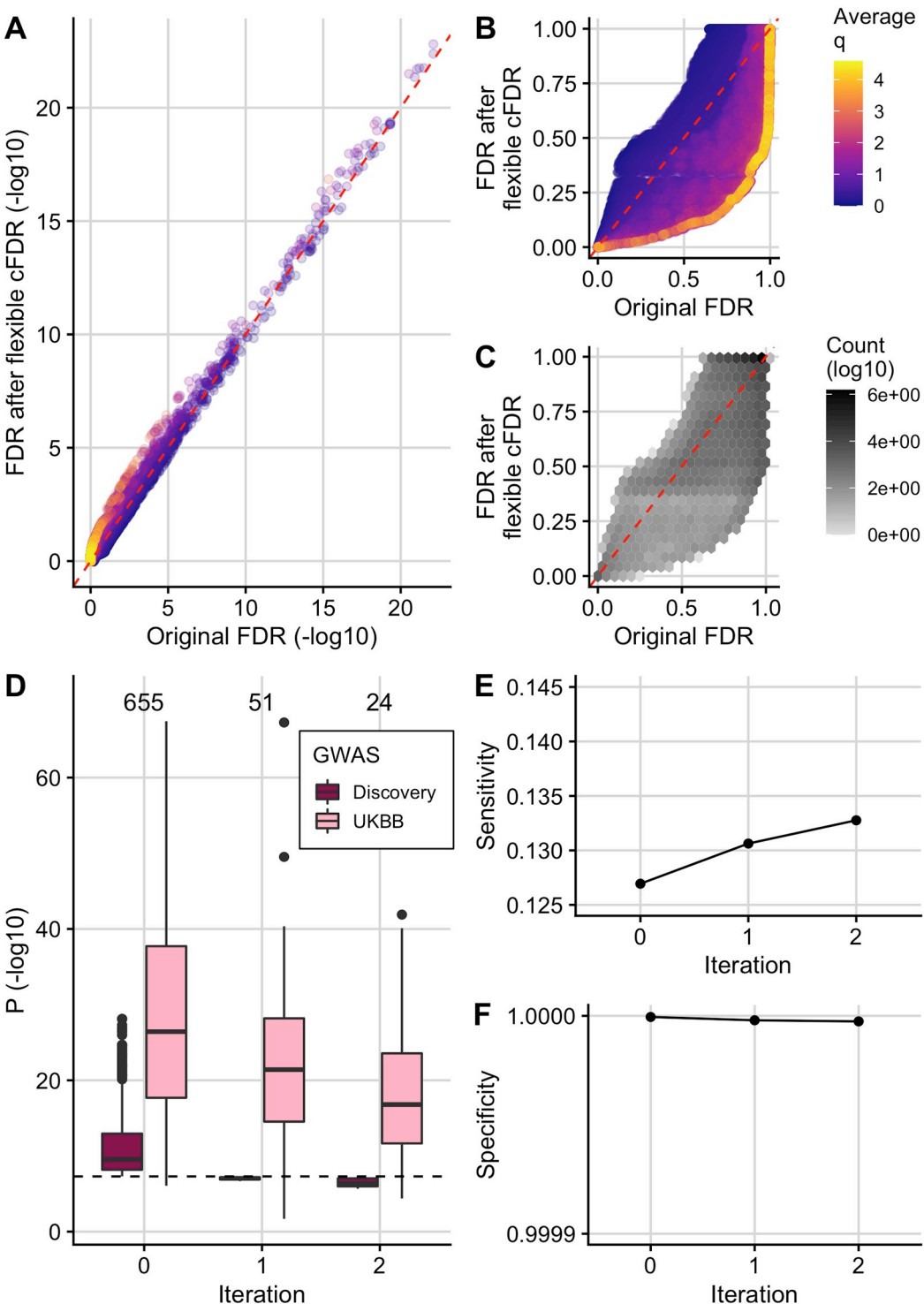

**Fig 3. Using Flexible cFDR to leverage H3K27ac data with asthma GWAS *p*-values.** (A) (-$\log_{10}$) FDR values after 2 iterations of Flexible cFDR leveraging H3K27ac counts in relevant cell types against raw (-$\log_{10}$) FDR values coloured by the average value of the auxiliary data across iterations. (B) As in A but non-log-transformed FDR values. (C) As in B but coloured by ($\log_{10}$) counts of data points in each hexbin. (D) Box plots of (-$\log_{10}$) *p*-values in the discovery GWAS and the UK Biobank data set for the 655 SNPs that were FDR significant in the original GWAS (Iteration 0), 51 SNPs that were newly FDR significant after iteration 1 of Flexible cFDR (leveraging average H3K27ac fold change values in lymphoid and CD56 cell types) and 24 SNPs that were newly FDR significant after iteration 2 of Flexible cFDR (subsequently leveraging average H3K27ac fold change values in lung tissue and CD14$^+$ cells). Black dashed line at genome-wide significance

($p = 5 \times 10^{-8}$). (E) Sensitivity proxy and (F) specificity proxy for the H3K27ac application results. Sensitivity proxy is calculated as the proportion of SNPs that are FDR significant in the UK Biobank data set that are also FDR significant in the original GWAS (iteration 0), after iteration 1 of Flexible cFDR or after iteration 2 of Flexible cFDR. Specificity is calculated as the proportion of SNPs that are not FDR significant in the UK Biobank data set that are also not FDR significant in the original GWAS (iteration 0), after iteration 1 of Flexible cFDR or after iteration 2 of Flexible cFDR.

fold change values in lymphoid and CD56 cell types (Fig 3D; Iteration 1), and 24 newly significant SNPs when subsequently leveraging average H3K27ac fold change values in lung tissue and monocytes (Fig 3D; Iteration 2). The maximum $p$-value for the 69 newly significant SNPs (6 SNPs newly significant after iteration 1 were no longer significant after iteration 2) in the discovery GWAS data set was $2.15 \times 10^{-6}$ and the maximum UK Biobank $p$-value for these SNPs was 0.02. The newly significant SNPs had relatively small estimated effect sizes (S16 Fig), implying that there may be many more regions associated with asthma with increasingly smaller effect sizes that are missed by current GWAS sample sizes.

As a proxy for sensitivity, we calculated the proportion of FDR significant SNPs in the UK Biobank data set that were also found to be FDR significant both before ("iteration 0") and after each iteration of Flexible cFDR. We found that the sensitivity increased from 0.127 to 0.131 after iteration 1 (leveraging average H3K27ac fold change values in lymphoid and CD56 cell types) and to 0.133 after iteration 2 (leveraging average H3K27ac fold change values in lung tissue and monocytes) (Fig 3E). As a proxy for specificity, we calculated the proportion of SNPs not FDR significant in the UK Biobank data set that were also not FDR significant both before ("iteration 0") and after each iteration of Flexible cFDR, finding that the specificity remained close to 1 ($\geq 0.9999975$) (Fig 3F). One could expect that the order of which the auxiliary data is iterated over may impact the results from Flexible cFDR. Reassuringly, in this application we found that the order of which we iterated over the auxiliary data had minimal impact on the results (S17 Fig).

At the locus level, 18 loci were FDR significant ($FDR \leq 0.000148249$) in the original asthma GWAS [47]. When used to leverage H3K27ac fold change values, Flexible cFDR identified 4 additional significant loci with index SNPs: rs9501077 (chr6:31167512), rs4148869 (chr6:32806576), rs9467715 (chr6:26341301) and rs167769 (chr12:57503775) (Fig 4 and Table 4). Three of the four (rs4148869, rs9467715 and rs167769) validated in the UK Biobank data set at Bonferroni corrected significance (for 4 tests the Bonferroni corrected significance threshold corresponding to $\alpha = 0.05$ is $0.05/4 = 0.0125$). One locus was found to be no longer FDR significant with index SNP rs12543811 (chr8:81278885).

SNPs rs9501077 and rs4148869 reside in the major histocompatibility complex (MHC) region of the genome, which is renowned for its strong long-range LD structures that make it difficult to dissect genetic architecture in this region. rs9501077 and rs4148869 are in linkage equilibrium ($r^2 = 0.001$), and are in very weak LD with the index SNP for the whole MHC region (rs9268969; $FDR = 7.35 \times 10^{-15}$; $r^2 = 0.005$ and $r^2 = 0.001$ respectively). rs9501077 ($p = 1.53 \times 10^{-7}$) has relatively high H3K27ac counts in asthma relevant cell types (mean percentile is 90th) and Flexible cFDR uses this extra disease-relevant information to increase the significance of this SNP beyond the significance threshold (FDR before Flexible cFDR = $3.99 \times 10^{-4}$, FDR after Flexible cFDR = $6.26 \times 10^{-5}$; Table 4). rs9501077 is found in the long non-coding RNA (lncRNA) gene, *HCG27* (HLA Complex Group 27), which has been linked to psoriasis [78], however the finding of a significant association with asthma for this SNP was not replicated in the UK Biobank data (UK Biobank $p = 0.020$).

SNP rs4148869 has very high H3K27ac fold change values in asthma relevant cell types (mean percentile is 99.6th) and so Flexible cFDR decreases the FDR value for this SNP from $9.28 \times 10^{-4}$ to $3.22 \times 10^{-5}$ when leveraging this auxiliary data (Table 4). This SNP is a 5' UTR

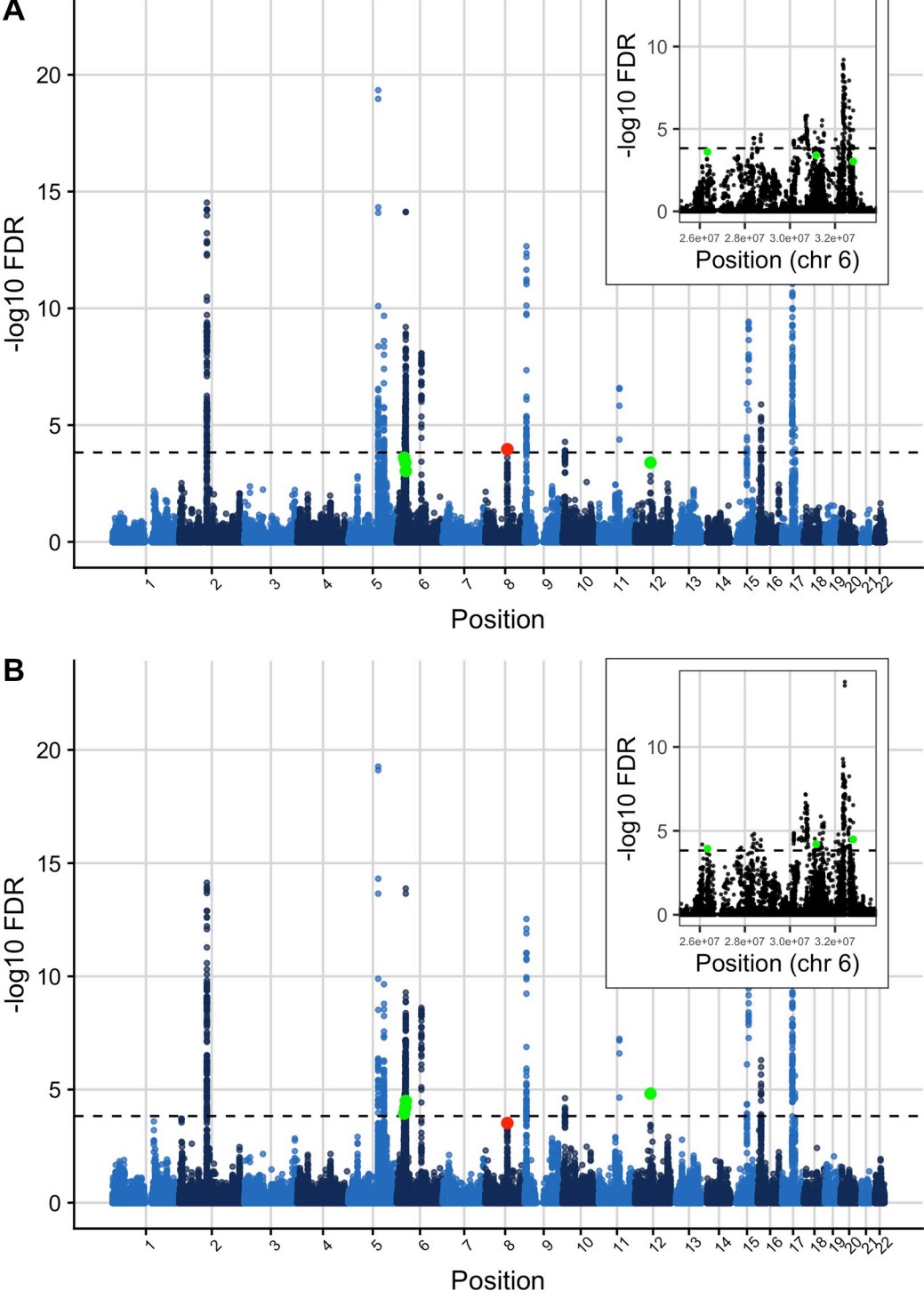

**Fig 4. Manhattan plot of FDR values before and after applying Flexible cFDR to leverage H3K27ac data with asthma GWAS *p*-values.** Manhattan plots of -$\log_{10}$ FDR values before (A) and after (B) applying Flexible cFDR leveraging H3K27ac counts in asthma relevant cell types. Points are coloured by chromosome and green points indicate the four index SNPs that are newly identified as FDR significant after Flexible cFDR (rs167769, rs9467715, rs9501077 and rs4148869) whilst the red point indicates the single index SNP that was newly identified as not FDR significant by Flexible cFDR (rs12543811). Black dashed line at FDR significance threshold [$-log_{10}(0.000148249)$].

**Table 4. Summary of newly significant asthma index SNPs when using Flexible cFDR to leverage H3K27ac data.** Details of index SNPs that became newly FDR significant (*FDR* < 0.000148249) after using Flexible cFDR to leverage H3K27ac fold change values with asthma GWAS *p*-values. Table contains the rsIDs (SNP), genomic positions (Chr: chromosome, BP: base pair), reference (Ref) and alternative (Alt) alleles, log ORs (beta), standard errors (SE) and *p*-values from the discovery GWAS, mean H3K27ac fold change values across asthma relevant cell types, *p*-values from UK Biobank and resultant *v*-values from Flexible cFDR. For the original *p*-values (and *v*-values), the corresponding FDR values are also given, calculated using the BH procedure.

| SNP | Chr | BP | Ref | Alt | beta | SE | H3K27ac percentile[a] | p | FDR (p) | p (UKBB) | v | FDR (v) |
|---|---|---|---|---|---|---|---|---|---|---|---|---|
| rs167769 | 12 | 57503775 | C | T | $7.87 \times 10^{-2}$ | $1.50 \times 10^{-2}$ | 98.4th | $1.55 \times 10^{-7}$ | $4.04 \times 10^{-4}$ | $4.69 \times 10^{-24}$ | $3.75 \times 10^{-9}$ | $1.51 \times 10^{-5}$ |
| rs9467715 | 6 | 26341301 | T | C | $-8.61 \times 10^{-2}$ | $1.61 \times 10^{-2}$ | 67.9th | $8.96 \times 10^{-8}$ | $2.49 \times 10^{-4}$ | $5.93 \times 10^{-4}$ | $3.83 \times 10^{-8}$ | $1.15 \times 10^{-4}$ |
| rs9501077 | 6 | 31167512 | A | G | $-8.06 \times 10^{-2}$ | $1.54 \times 10^{-2}$ | 90.5th | $1.53 \times 10^{-7}$ | $3.99 \times 10^{-4}$ | $2.01 \times 10^{-2}$ | $1.91 \times 10^{-8}$ | $6.26 \times 10^{-5}$ |
| rs4148869 | 6 | 32806576 | C | T | $7.03 \times 10^{-2}$ | $1.39 \times 10^{-2}$ | 99.6th | $4.03 \times 10^{-7}$ | $9.28 \times 10^{-4}$ | $1.49 \times 10^{-15}$ | $9.07 \times 10^{-9}$ | $3.22 \times 10^{-5}$ |

[a] Mean H3K27ac percentile across cell types.

variant in the *TAP2* gene. The protein TAP2 assembles with TAP1 to form a transporter associated with antigen processing (TAP) complex. The TAP complex transports foreign peptides to the endoplasmic reticulum where they attach to MHC class I proteins which in turn are trafficked to the surface of the cell for antigen presentation to initiate an immune response [79]. Studies have found *TAP2* to be associated with various immune-related disorders, including autoimmune thyroiditis and type 1 diabetes [80, 81], and pulmonary tuberculosis in Iranian populations [82]. Recently, Ma and colleagues [83] identified three cis-regulatory eSNPS for *TAP2* as candidates for childhood-onset asthma risk (rs9267798, rs4148882 and rs241456). One of these (rs4148882) is present in the asthma GWAS data set used for our analysis (*FDR* = 0.12) and is in weak LD with rs4148869 ($r^2$ = 0.4).

SNP rs9467715 is a regulatory region variant with a raw FDR value that is very nearly significant in the original GWAS (*FDR* = $2.49 \times 10^{-4}$ compared with FDR threshold of $1.48 \times 10^{-4}$ used to call significant SNPs). This SNP has moderate H3K27ac fold change values in asthma relevant cell types (mean percentile is 67.9th) so that when these are leveraged using Flexible cFDR, the SNP is just pushed past the FDR significance threshold (FDR after Flexible cFDR = $1.15 \times 10^{-4}$; Table 4).

SNP rs167769 has a borderline FDR value in the original GWAS discovery data set (*FDR* = $4.04 \times 10^{-4}$) but was found to be significant in the multi-ancestry analysis in the same manuscript (*FDR* = $1.61 \times 10^{-5}$) [47]. This SNP has very high H3K27ac fold change values in asthma relevant cell types (mean percentile is 98.4th) and Flexible cFDR decreases the FDR value for this SNP to $1.51 \times 10^{-5}$ when leveraging this auxiliary data (Table 4). rs167769 is an intron variant in *STAT6*, a gene that is activated by cytokines IL-4 and IL-13 [84, 85] to initiate a Th2 response and ultimately inhibit transcribing of innate immune response genes [86, 87]. Transgenic mice over-expressing constitutively active *STAT6* in T cells are predisposed towards Th2 responses and allergic inflammation [88, 89] whilst *STAT6*-knockout mice are protected from allergic pulmonary manifestations [90]. Accordingly, rs167769 is strongly associated with *STAT6* expression in the blood [91–93] and lungs [94] and is associated with increased risk of childhood atopic dermatitis [95, 96], which often progresses to allergic airways diseases such as asthma in adulthood. No genetic variants in the *STAT6* gene region (chr12:57489187–57525922) were identified as significant in the original GWAS, and only rs167769 was identified as significant after leveraging H3K27ac data using Flexible cFDR (S18 Fig).

One significant index SNP was no longer significant after applying Flexible cFDR. rs12543811 is located between genes *TPD52* and *ZBTB10* and has moderate H3K27ac fold change values in asthma relevant cell types (mean percentile is 52th). This SNP only just

exceeds the FDR significance threshold in the original GWAS (FDR = $1.08 \times 10^{-4}$ compared with FDR threshold of $1.48 \times 10^{-4}$ used to call significant SNPs) but by leveraging its H3K27ac fold change values using Flexible cFDR, the resultant FDR value is just below the significance threshold (FDR after Flexible cFDR = $3.04 \times 10^{-4}$; S19 Fig). This SNP is in strong LD with rs7009110 ($r^2 = 0.79$) which has previous been associated with asthma plus hay fever but not with asthma alone [97]. Conditional analyses show that these two SNPs represent the same signal which is likely to be associated with allergic asthma [47]. rs12543811 was found to be significant in the UK Biobank data (UK Biobank $p = 1.42 \times 10^{-19}$).

## Comparison with existing methods

**Boca and Leek's FDR regression.** We compared the results from Flexible cFDR when leveraging cell-type specific ChIP-seq data to those from BL when leveraging the exact same auxiliary data (Fig 5). The estimated probabilities that each SNP was null (not associated), which are calculated as an intermediate step in the method, ranged from 0.746 to 1 and were negatively correlated with H3K27ac fold change values in asthma relevant cell types (S20 Fig). In total, BL identified five SNPs as newly FDR significant, which replicate in the

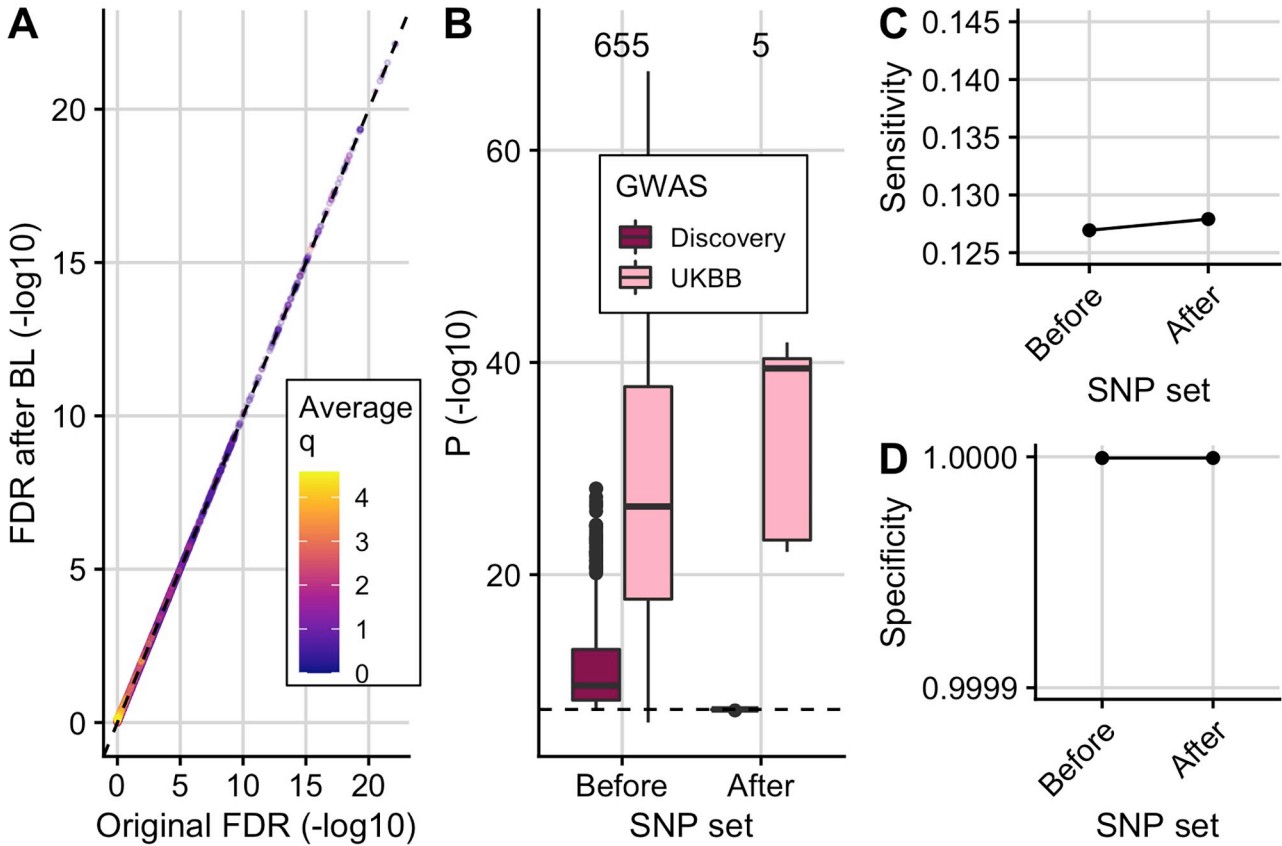

**Fig 5. Using Boca and Leek's FDR regression to leverage H3K27ac data with asthma GWAS $p$-values.** (A) (-$\log_{10}$) adjusted $p$-values from BL against raw (-$\log_{10}$) FDR values coloured by average value of $q$ (H3K27ac fold change value). (B) Box plots of (-$\log_{10}$) $p$-values in the discovery GWAS data set and the UK Biobank data set for the 655 SNPs that were FDR significant in the original GWAS ("before") and 5 newly significant SNPs after applying BL ("after"). Black dashed line at genome-wide significance threshold ($5 \times 10^{-8}$). (C) Sensitivity and (D) specificity proxies for the results. Sensitivity proxy is calculated as the proportion of SNPs that are FDR significant in the UK Biobank data set that are also FDR significant in the original GWAS or after applying BL. Specificity is calculated as the proportion of SNPs that are not FDR significant in the UK Biobank data set that are also not FDR significant in the original GWAS or after BL.

UK Biobank validation data set (rs4705950 UK Biobank $p = 7.2 \times 10^{-23}$, rs9268831 UK Biobank $p = 1.3 \times 10^{-42}$, rs17533090 UK Biobank $p = 4.4 \times 10^{-41}$, rs1871665 UK Biobank $p = 5.5 \times 10^{-24}$ and rs16924428 UK Biobank $p = 3.6 \times 10^{-40}$) (Fig 5B). These SNPs are a subset of the 69 newly significant SNPs identified by Flexible cFDR, except for rs16924428 which has very low H3K27ac fold change values in asthma relevant cell types (average value = 0.087). The sensitivity increased slightly from 0.127 to 0.128 after applying BL (compared to 0.133 after Flexible cFDR) (Fig 5C) and the specificity remained stable at 0.9999995 (Fig 5D). No SNPs were FDR significant in the discovery data set and no longer FDR significant after applying BL, and no new loci were found to be newly FDR significant (or newly not FDR significant).

**FINDOR.** We next compared results from Flexible cFDR when leveraging cell-type specific ChIP-seq data to those from FINDOR which leverages a wider range of non-cell-type-specific functional annotations, and have made all of the results publicly available at https://doi.org/10.5281/zenodo.5554628. FINDOR identified 119 newly FDR significant SNPs which had a median $p$-value of $4.44 \times 10^{-15}$ in the UK Biobank validation data, but the maximum UK Biobank $p$-value for these 119 newly significant SNPs was 0.98 (Fig 6A and 6B). The proportion of FDR significant SNPs in the UK Biobank data set that were also FDR significant in the discovery GWAS data set increased from 0.127 to 0.146 (compared to 0.128 after BL and 0.133 after Flexible cFDR) (Fig 6C) and the specificity remained $\geq 0.99999$ (Fig 6D). The increase in sensitivity from FINDOR is greater than that of Flexible cFDR and BL, which may reflect the information gain in leveraging 96 annotations rather than a single histone mark.

At the locus level, FINDOR identified two newly FDR significant index SNPs: rs13018263 (chr2:103092270; original $FDR = 6.79 \times 10^{-4}$, new $FDR = 10^{-4}$) and rs9501077 (chr6:31167512; original $FDR = 3.99 \times 10^{-4}$, new $FDR = 4.86 \times 10^{-5}$) (Fig 6E and 6F). SNP rs13018263 is an intronic variant in *SLC9A4* and is strongly significant in the UK Biobank validation data set ($p = 4.78 \times 10^{-31}$). Ferreira and colleagues [98] highlighted rs13018263 as a potential eQTL for *IL18RAP*, a gene which is involved in IL-18 signalling which in turn mediates Th1 responses [99], and is situated just upstream of *SLC9A4*. Genetic variants in *IL18RAP* are associated with many immune-mediated diseases, including atopic dermatitis [100] and type 1 diabetes [101]. Interestingly, although different auxiliary data was leveraged using Flexible cFDR and FINDOR in our analyses, both methods found index SNP rs9501077 to be newly significant, but this SNP did not validate in the UK Biobank data (UK Biobank $p = 0.020$).

Two additional index SNPs were found to be no longer significant after re-weighting by FINDOR, rs2589561 (chr10:9046645; original $FDR = 5.25 \times 10^{-5}$, new $FDR = 3.06 \times 10^{-3}$) and rs17637472 (chr17:47461433; original $FDR = 1.42 \times 10^{-5}$, new $FDR = 9.42 \times 10^{-4}$), however both of these SNPs were strongly significant in the UK Biobank validation data set ($p = 2.09 \times 10^{-29}$ and $p = 1.75 \times 10^{-14}$ respectively).

SNP rs2589561 is a gene desert that is 929kb from *GATA3*, a transcription factor of the Th2 pathway which mediates the immune response to allergens [47, 102]. Hi-C data in hematopoietic cells showed that two proxies of rs2589561 ($r^2 > 0.9$) are located in a region that interacts with the *GATA3* promoter in CD4+ T cells [103], suggesting that rs2589561 could function as a distal regulator of *GATA3* in this asthma relevant cell type. rs2589561 has relatively high H3K27ac fold change values in the asthma relevant cell types leveraged by Flexible cFDR (mean percentile is 85th) and Flexible cFDR decreased the FDR value from $5.25 \times 10^{-5}$ to $2.41 \times 10^{-5}$.

SNP rs17637472 is a strong cis-eQTL for *GNGT2* in blood [91–93, 104], a gene whose protein product is involved in NF-$\kappa$B activation [105]. This SNP has moderate H3K27ac fold change values in relevant cell types (mean percentile is 62th) and the FDR values for this SNP

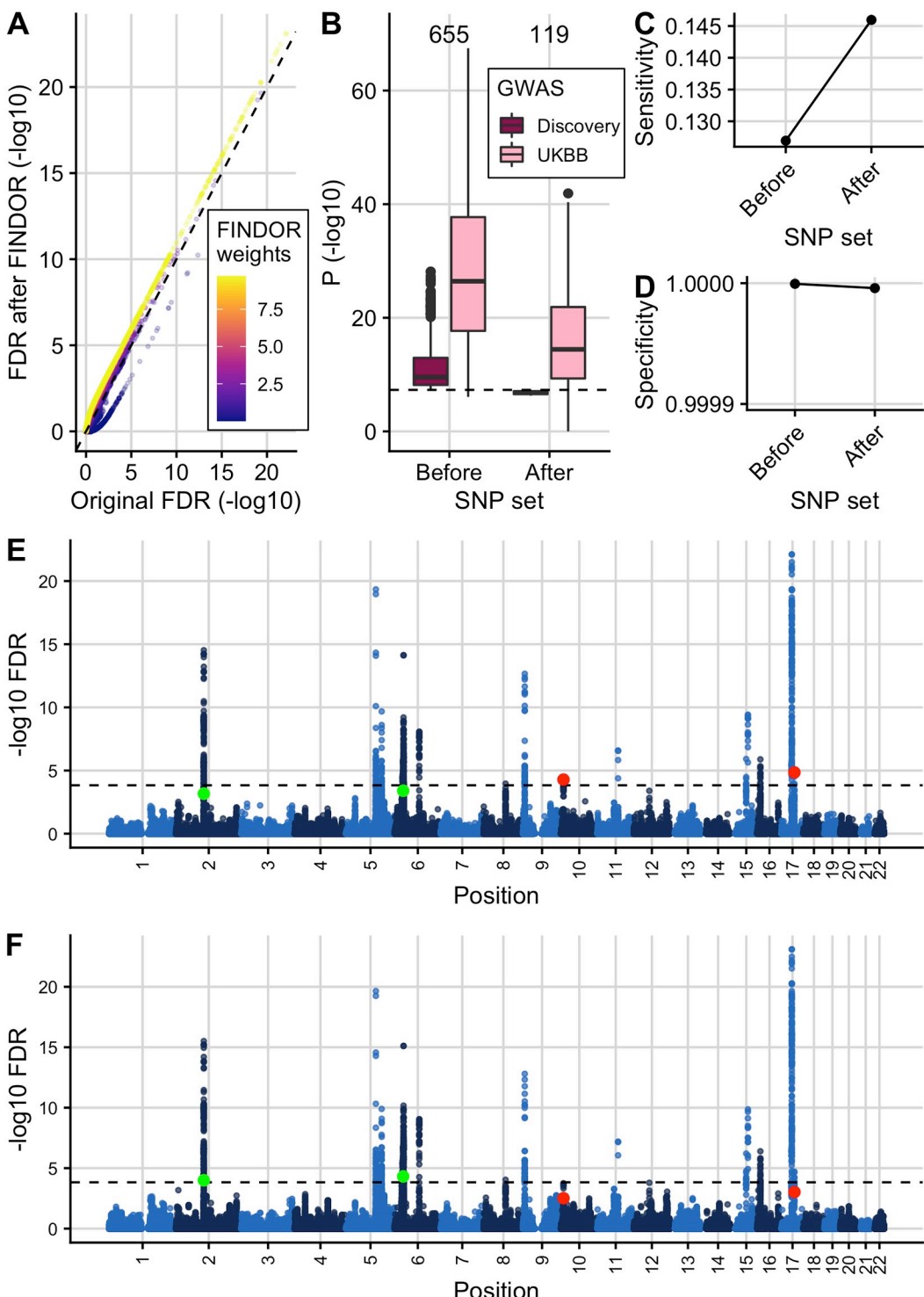

**Fig 6. Results from FINDOR re-weighting of asthma GWAS *p*-values leveraging 96 baseline-LD model annotations.** (A) (-log$_{10}$) FDR values from FINDOR against (-log$_{10}$) original FDR values coloured by FINDOR weights. (B) Box plots of (-log$_{10}$) *p*-values in the discovery GWAS data set and the UK Biobank data set for the 655 SNPs that were FDR significant in the original GWAS ("before") and 119 newly significant SNPs after re-weighting using FINDOR ("after"). Black dashed line at genome-wide significance threshold ($5 \times 10^{-8}$). (C) Sensitivity and (D) specificity proxies for the FINDOR results. Sensitivity proxy is calculated as the proportion of SNPs that are FDR significant in the UK Biobank data set that are also FDR significant in the original GWAS or after *p*-value re-weighting using FINDOR. Specificity is calculated as the proportion of SNPs that are not FDR significant in the UK Biobank data set that are also not FDR significant in the original

GWAS or after $p$-value re-weighting using FINDOR. Manhattan plots of FDR values before (E) and after (F) re-weighting by FINDOR. Green points indicate the two index SNPs that are newly identified as FDR significant by FINDOR [rs13018263 (chr2:103092270) and rs9501077 (chr6:31167512)]. Red points indicate the two index SNPs that are newly identified as not FDR significant by FINDOR [rs2589561 (chr10:9046645) and rs17637472 (chr17:47461433)]. Black dashed line at FDR significance threshold [$-log_{10}(0.000148249)$].

were similar both before and after using Flexible cFDR to leverage the H3K27ac data (original $FDR = 1.42 \times 10^{-5}$, new $FDR = 1.46 \times 10^{-5}$).

## Discussion

Developments in molecular biology have enabled researchers to decipher the functional effects of various genomic signatures. We are now in a position to prioritise sequence variants associated with various phenotypes not just by their genetic association statistics but also based on our biological understanding of their functional role. Originally designed for the specific purpose of leveraging test statistics from genetically related traits, we have extended the cFDR framework [44] to support auxiliary data from arbitrary continuous distributions. Our extension, Flexible cFDR, provides a statistically robust framework to leverage functional genomic data with genetic association statistics to boost power for GWAS discovery.

We compared the performance of Flexible cFDR to that of four comparator methods which have previously been shown to outperform other approaches [15, 48]: GenoWAP, IHW, BL and FINDOR. We also tried to compare our method to AdaPT [10], but this approach uses a $p$-value masking procedure which takes many iterations of optimisation and can be computationally expensive [106]. We found AdaPT to be too computationally demanding for large-scale GWAS data and previous studies suggest that a SNP pre-filtering stage is required [37]. Of the methods considered, we found that only BL was as versatile as Flexible cFDR. Specifically, IHW currently only supports univariate covariates and, unlike Flexible cFDR, cannot be applied iteratively to leverage multi-dimensional covariates. In GenoWAP, the prior probabilities used in the model are calculated as the mean GenoCanyon score (or tissue-specific GenoSkyline [107] or GenoSkyline-Plus [108] score) of the surrounding 10, 000 base pairs, thereby restricting its utility to leveraging only these scores (which we found were unlikely to capture enough disease relevant information to substantially alter conclusions from a study). Whilst in FINDOR, SNPs are binned based on how well they tag heritability enriched categories and this requires the estimation of $\chi^2$ statistics (i.e., tagged variance) for each SNP using a range of functional annotations, which are generally those in the baselineLD model [27]. Users are thus required to run LD-score regression prior to running FINDOR, and this two-step approach may limit the accessibility of the method. Although BL was as versatile as Flexible cFDR, we found that it failed to control the FDR in some simulations and was less powerful than Flexible cFDR in a simulation-based analysis. Whilst FINDOR was shown to be the most powerful method, this may reflect the information gain in leveraging 96 annotations rather than a single histone mark. This emphasises the importance of being able to iterate over different auxiliary measures, and suggests that a fruitful area of extension for cFDR will be to increase the robustness of FDR control for dependent $q$ across multiple iterations. A related extension would be to formally assess robustness of the cFDR approach to the order of which the auxiliary data is iterated over.

Flexible cFDR has several key advantages over competing methods. It does not bin variables and does not rely on subjective thresholding or normalised weighting schemes, which hinder many of the existing methods [3–6, 9, 14, 15, 25]. Moreover, Flexible cFDR outputs quantities analogous to $p$-values which can be used directly in any error-rate controlling

procedure, and which also permit iteration to support multi-dimensional covariate data. Whilst LD between SNPs is often a concern (e.g. because methods such as KDE assume independence between observations), we fit the KDE to a subset of LD-independent SNPs but then generate $v$-values for the full set of SNPs, thereby benefiting from computational efficiency but also facilitating downstream analyses which typically require the full set of SNPs, such as fine-mapping or meta-analysis. LD means that the $v$-values will be positively correlated, so we appeal to the established robustness of the BH FDR estimation to positive dependency [109].

Whilst larger case and control cohort sizes will boost statistical power for GWAS discovery, incorporating functional data provides an additional layer of biological evidence that an increase in sample sizes alone cannot provide. There are also instances in the rare disease domain where case sample sizes are restricted by the number of cases available for recruitment (for example in primary immunodeficiency disorder [110]), and our method has potential utility in these instances as it provides an alternative approach to increase statistical power. The choice of functional data to use may be guided by prior knowledge, or in a data driven manner using a method such as GARFIELD [111] to quantify the enrichment of GWAS signals in different functional marks. Moreover, our method intrinsically evaluates the relevance of the auxiliary data by comparing the joint probability density of the test statistics and the auxiliary data to the joint density assuming independence, and can therefore be used to inform researchers of relevant functional signatures and cell types.

Our manuscript describes four key advances enabling the extension of the cFDR framework to the functional genomics setting. Firstly, we derive an estimator based on a 2-dimensional KDE of the bivariate distribution rather than empirical estimates, making our method considerably faster than earlier empirical approaches. Secondly, the cFDR framework estimates $q|H_0^p$ with the relatively coarse approximation $q|p > 1/2$. In contrast, Flexible cFDR utilises the local false discovery rate in its estimator for $q|H_0^p$. The local false discovery rate conditions the probability of a null hypothesis on the point value of its $p$-value, and its use allows for finer-grained estimation of $q|H_0^p$. Thirdly, we remove the assumption that $q|H_0^p$ can be transformed to a mixture of centered normals, and instead integrate over the previously estimated KDE, which relaxes the distributional assumptions placed on the auxiliary data. Finally, Flexible cFDR is supported by user-oriented software documented on an easy-to-navigate website (https://annahutch.github.io/fcfdr/). The website features several fully reproducible vignettes which illustrate how the method can be applied to a particular data set at the desired level of error control. We hope this support will make Flexible cFDR accessible to a wider range of researchers.

One can see that the performance of Flexible cFDR depends on how well the KDE fits the data. Usual concerns about KDE apply, including that fits may be poor if there are regions with very sparse data. The auxiliary data can be transformed to improve the KDE fitting procedure, as in application 2 where we log-transformed the raw fold change values to avoid long tails. The optimal scale for the auxiliary data is likely to depend on the relationship between the principal $p$-values and the auxiliary data, and is not something we have explored here, but as usual, data visualisation is likely to be helpful to confirm that the scale for the auxiliary data is sensible. By default, the Flexible cFDR software returns a plot of the fitted 2D KDE and the estimated density of the auxiliary data overlaid onto the real data values, enabling users to visually examine the fit of the KDE to their data. Investigating the KDE fitting procedure, including the suitability of the Gaussian kernel, is an interesting area for future research.

Estimation of both FDR and local false discovery rates (often denoted "fdr") [32] require estimation of a ratio. In the case of FDR these are cumulative distribution functions (CDF),

and in the case of the fdr these are probability density functions (PDF). The simple approach is to take a ratio of two separate estimates of the numerator and denominator. In the case of a 2d fdr proposed by Ploner et al. [34], this approach was found to be numerically unstable, because the sparsity of the data across 2 dimensions means the uncertainty attached to the estimate of the denominator in particular may be large. Ploner et al. [34] proposed a solution based on binomial regression, adapting the Poisson regression method used by Efron [32] to estimate the denominator in the fdr ratio. Despite depending on a 2 dimensional density, we do not use a smoothing estimator for cFDR, yet it still performs well. This is because CDF estimates are typically more stable than PDF estimates at any given point (apart from towards the lower limits of the data), and is one of the attractive ideas for using cumulative rather than probability density functions.

Our method has several limitations. Firstly, if applying Flexible cFDR iteratively then it is important that each iteration considers new information. That is, care must be taken to ensure that the auxiliary data to be leveraged iteratively is capturing distinct disease-relevant features to prevent multiple adjustment using the same auxiliary data. The definition of "distinct disease-relevant features" to leverage is at the user's discretion and sparks an interesting philosophical discussion. For example, leveraging data iteratively from various genomic assays measuring the same genomic feature at different resolutions may be deemed invalid for some researchers but valid for others, since if the mark is repeatedly identified by different assays then it is more likely to be reliably present. Whilst we show that our method is robust to minor departures from $q_i^{(k)} \perp\!\!\!\perp q_i^{(l)}|H_0^p$, this does not extend to strongly related $q$. We would argue that the conservative approach would be to average over correlated auxiliary data, to ensure that the $q$ vectors are not strongly correlated.

Secondly, Flexible cFDR requires that the auxiliary data to be leveraged is continuous. This means that the approach cannot currently be used to leverage functional genomic data that yields discrete or binary values, such as PHRED scores, whether SNPs are synonymous or non-synonymous or whether they reside in coding regions of the genome. A fruitful contribution to the field would be to extend the cFDR approach to support discrete or binary data, thus increasing applicability.

Thirdly, the cFDR framework assumes a positive stochastically monotonic relationship between the test statistics and the auxiliary data: specifically, low $p$-values are enriched for low values in the auxiliary data. Our method automatically calculates the correlation between $p$ and $q$ and if this is negative then the auxiliary data is transformed to $q := -q$. However, if the relationship is non-monotonic (for example low $p$-values are enriched for both very low and very high values in the auxiliary data) then the cFDR framework cannot simultaneously shrink $v$-values for these two extremes. This non-monotonic relationship is unlikely when leveraging single functional genomic marks, but may occur if, for example, multiple marks were decomposed via PCA. We therefore recommend that users use the `corr_plot` and `stratified_qqplot` functions in the `fcfdr` R package to visualise the relationship between the relationship between the two data types. Note that this restriction could be removed if we used density instead of distribution functions, and worked at the level of local false discovery rates [32] as described earlier, but this would in turn reduce the robustness our method has to data sparsity in the $(p, q)$ plane.

Finally, in our asthma application we only leveraged data for a single histone modification across various cell types. Additional data measuring other histone modifications (e.g. repressive marks) could also be leveraged to further increase power.

Overall, we anticipate that Flexible cFDR will be a valuable tool to leverage functional genomic data with GWAS test statistics to boost power for GWAS discovery.

## Supporting information

**S1 Fig. Demonstration of the left-censoring procedure.** Plots showing how many data points are in each grid space of the auxiliary data, $q$, over the support of the KDE for an example data set. (A) shows the full support of the KDE and (B) is zoomed in to the left tail. Black dashed line at $y = 50$ which is the default value of the `gridp` parameter in the `fcfdr::flexible_cfdr` function. Data points falling in grid spaces with fewer than 50 data points (those to the left of the blue dashed line) are left-censored, meaning that their value is replaced by the value of the left bound of the first grid space containing more than 50 data points. In practise, very few data points are left-censored.
(TIF)

**S2 Fig. Illustration of the spline correction procedure.** A spline with 5 knots is fitted to $log_{10}(v/p)$ against $q$ using the `bigsplines` R package (https://cran.r-project.org/web/packages/bigsplines/index.html) for an example data set. The distance between each data point and the fitted spline is calculated. If this distance is greater than the value of the `dist_thr` parameter in the `fcfdr::flexible_cfdr` function (default value is 0.5), then the data point is mapped back to the spline and the corresponding $v$-value is recalculated using the fitted spline. In this example, the red line shows the fitted spline and the grey triangular points are mapped back to the spline to generate new $v$-values.
(TIF)

**S3 Fig. Histograms of auxiliary data leveraged in simulation analysis.** (A) Example data leveraged in simulation A (simulated from standard uniform distribution). (B) Example data leveraged in simulation B (simulated $p$-values for related traits). (C) Example data leveraged in simulations C, D and E (simulated from a mixture normal distribution).
(TIF)

**S4 Fig. Choosing an FDR threshold corresponding to the genome-wide significance $p$-value threshold in the simulation analysis.** Histogram of the maximum FDR-adjusted $p$-value (using BH method) amongst SNPs with $p \leq 5 \times 10^{-8}$ in the simulation analysis. Red dashed line at the selected FDR threshold of $5 \times 10^{-6}$.
(TIF)

**S5 Fig. Histograms from GenoCanyon application.** (A) Histogram of GenoCanyon scores for SNPs in the asthma GWAS data set. (B) Histogram of estimated pi0 values from BL.
(TIF)

**S6 Fig. Heatmap of the correlations between H3K27ac fold change values amongst asthma relevant cell types.**
(TIF)

**S7 Fig. Histograms of auxiliary data used in H3K27ac application.** (A) q1 is the average of (log transformed) H3K27ac fold change values in lymphoid and CD56 cell types (B) q2 is the average of (log transformed) H3K27ac fold change values in lung tissue and CD14+ cells.
(TIF)

**S8 Fig. Simulation results assessing the sensitivity when increasing the $r^2$ threshold used to call associated SNPs.** Mean +/- standard error for the sensitivity of FDR values from empirical and Flexible cFDR when iterating over independent (A; "simulation A") and dependent (B; "simulation B") auxiliary data that is bounded by [0, 1]. Panels C and D show the results from Flexible cFDR when iterating over independent (C; "simulation C") and dependent (D; "simulation D") auxiliary data simulated from bimodal mixture normal distributions. BL refers to

results when using Boca and Leek's FDR regression to leverage the 5-dimensional covariate data. Iteration 0 corresponds to the original FDR values. Our sensitivity proxy is calculated as the proportion of SNPs with $r^2 \geq X$ with a causal variant ("truly associated"), that were detected with a FDR value less than $5 \times 10^{-6}$, where results are faceted for $X = 0.8, 0.9, 0.95$. Results were averaged across 100 simulations.
(TIF)

**S9 Fig. Simulation results assessing the specificity when increasing the $r^2$ threshold used to call not-associated SNPs.** Mean +/- standard error for the specificity of FDR values from empirical and Flexible cFDR when iterating over independent (A; "simulation A") and dependent (B; "simulation B") auxiliary data that is bounded by [0, 1]. Panels C and D show the results from Flexible cFDR when iterating over independent (C; "simulation C") and dependent (D; "simulation D") auxiliary data simulated from bimodal mixture normal distributions. BL refers to results when using Boca and Leek's FDR regression to leverage the 5-dimensional covariate data. Iteration 0 corresponds to the original FDR values. Our specificity proxy is calculated as the proportion of SNPs with $r^2 \leq X$ with all the causal variants ("truly not-associated"), that were not detected with a FDR value less than $5 \times 10^{-6}$, where results are faceted for $X = 0.01, 0.05$. Results were averaged across 100 simulations.
(TIF)

**S10 Fig. Simulation results for using Flexible cFDR to iteratively leverage exactly the same auxiliary data.** Mean +/- standard error for the sensitivity (A) specificity (B) and FDR (C) of FDR values from Flexible cFDR when iterating over the same dependent auxiliary data ("simulation E"). Iteration 0 corresponds to the original FDR values. Our sensitivity proxy is calculated as the proportion of SNPs with $r^2 \geq 0.8$ with a causal variant ("truly associated"), that were detected with a FDR value less than $5 \times 10^{-6}$. Our specificity proxy is calculated as the proportion of SNPs with $r^2 \leq 0.01$ with all the causal variants ("truly not-associated"), that were not detected with a FDR value less than $5 \times 10^{-6}$. Our FDR proxy is calculated as the proportion of SNPs that were detected with a FDR value less than 0.05, that had $r^2 \leq 0.01$ with all the causal variants ("truly not-associated") (we raised $\alpha$ to 0.05 in order to assess FDR control within a manageable number of simulations). Results were averaged across 1000 simulations.
(TIF)

**S11 Fig. Stratified Q-Q plot of empirical -$\log_{10}$ GWAS $p$-values for asthma against theoretical values stratified by GenoCanyon scores.** The values that were used to threshold the GenoCanyon scores are the quantiles of the distribution (0.020 is the 0.25 quantile, 0.204 is the 0.5 quantile, 0.731 is the 0.75 quantile and 1 is the maximum value).
(TIF)

**S12 Fig. Results when using Flexible cFDR to leverage GenoCanyon scores with asthma GWAS $p$-values.** FDR values after using Flexible cFDR to leverage GenoCanyon scores with asthma GWAS $p$-values against raw FDR values coloured by GenoCanyon score.
(TIF)

**S13 Fig. Results when using BL to leverage GenoCanyon scores with asthma GWAS $p$-values.** Adjusted $p$-values from BL when leveraging GenoCanyon scores with asthma GWAS $p$-values against raw FDR values coloured by GenoCanyon score.
(TIF)

**S14 Fig. Heatmap of correlations between log-transformed asthma GWAS $p$-values and the summarised H3K27ac fold change values leveraged by Flexible cFDR.** q1 is the average of (log transformed) H3K27ac fold change values in lymphoid and CD56 cell types. q2 is the

average of (log transformed) H3K27ac fold change values in lung tissue and CD14+ cells.
(TIF)

**S15 Fig. Stratified Q-Q plot of empirical -$\log_{10}$ GWAS *p*-values for asthma against theoretical values stratified by average H3K27ac fold change values in asthma relevant cell types.** The values that were used to threshold *q* (average H3K27ac fold change values) are the quantiles of the distribution (0.174 is the 0.25 quantile, 0.271 is the 0.5 quantile, 0.419 is the 0.75 quantile and 4.583 is the maximum value).
(TIF)

**S16 Fig. Effect sizes of significant SNPs.** Absolute estimated effect sizes ($|\beta|$; log OR) +/ $-1.96$ $\times$ *SE* of SNPs significantly associated (*FDR* $\leq 0.000148249$) with asthma in the original discovery GWAS data set ("iteration 0") and those newly significant after iteration 1 and 2 of cFDR.
(TIF)

**S17 Fig. Switching the order of iteration.** (-$\log_{10}$) *v*-values after 2 iterations of Flexible cFDR leveraging H3K27ac data when iterating over q2 and then q1 against (-$\log_{10}$) *v*-values when iterating over q1 then q2.
(TIF)

**S18 Fig. Manhattan plots for genomic region containing the *STAT6* gene.** Manhattan plots of FDR values before (A) and after (B) applying Flexible cFDR for the region containing the *STAT6* gene (chr12:57489187–57525922). Black dashed line at FDR significant threshold. Red SNP is rs167769 (index SNP).
(TIF)

**S19 Fig. Manhattan plots for region containing the index SNP rs12543811 that is no longer FDR significant after applying Flexible cFDR.** Manhattan plots of FDR values before (A) and after (B) applying Flexible cFDR for the region (chr8:81100000–81500000) containing index SNP rs12543811 (chr6:81278885) that is no longer FDR significant when applying Flexible cFDR.
(TIF)

**S20 Fig. Estimated pi0 values in BL when leveraging H3K27ac fold change values in relevant cell types with asthma GWAS *p*-values.** (A) Average H3K27ac fold change values in asthma relevant cell types (*q*) against estimated probabilities that the null hypothesis is true ('pi0') (B) Histogram of pi0 values for all 1,968,651 SNPs.
(TIF)

## Author Contributions

**Conceptualization:** Anna Hutchinson, Chris Wallace.

**Data curation:** Anna Hutchinson, Guillermo Reales.

**Formal analysis:** Anna Hutchinson, Thomas Willis, Chris Wallace.

**Funding acquisition:** Anna Hutchinson, Chris Wallace.

**Investigation:** Anna Hutchinson, Thomas Willis, Chris Wallace.

**Methodology:** Anna Hutchinson, Chris Wallace.

**Resources:** Anna Hutchinson, Chris Wallace.

**Software:** Anna Hutchinson, Thomas Willis, Chris Wallace.

**Supervision:** Chris Wallace.

**Validation:** Anna Hutchinson, Thomas Willis, Chris Wallace.

**Visualization:** Anna Hutchinson, Thomas Willis, Chris Wallace.

**Writing – original draft:** Anna Hutchinson, Guillermo Reales, Thomas Willis, Chris Wallace.

**Writing – review & editing:** Anna Hutchinson, Guillermo Reales, Thomas Willis, Chris Wallace.

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
