## [Decision Letter · Decision Letter 0]

5 Sep 2021

Dear Dr Hutchinson,

Thank you very much for submitting your Research Article entitled 'Leveraging auxiliary data from arbitrary distributions to boost GWAS discovery with Flexible cFDR' to PLOS Genetics.

The manuscript was evaluated at the editorial level and by independent peer reviewers. Two reviewers have made comments and raised concerns and that we would like you to address in a revised version. Technically the issues raised are substantial enough that we have made a "major revision" decision, but we believe that you should be able to respond to them without great difficulty.

If you decide to revise the manuscript for further consideration at PLOS Genetics, please aim to resubmit within the next 60 days, unless it will take extra time to address the concerns of the reviewers, in which case we would appreciate an expected resubmission date by email to plosgenetics@plos.org.

[LINK]

We are sorry that we cannot be more positive about your manuscript at this stage. Please do not hesitate to contact us if you have any concerns or questions.

Yours sincerely,

Jingjing Yang, Ph.D.

Guest Editor

PLOS Genetics

David Balding

Section Editor: Methods

PLOS Genetics

Reviewer's Responses to Questions

Reviewer #1: The authors have done a good job of addressing comments from the previous round of reviews.

Reviewer #2: The manuscript, "Leveraging auxiliary data from arbitrary distributions to boost GWAS discovery with Flexible cFDR", describes an FDR approach for GWAS in the presence of secondary data-sets such as GWAS summary statistics of related traits or variant-level annotation scores. The methods is based on a widely accepted concept, the conditional false discovery rate (cFDR). It overcomes a major limition of previous work that the secondary data-set must follow a mixture Gaussian distribution, thus enabling the application of the proposed approach to integrating a variety of secondary data-sets for GWAS on a complex trait, in particular functional genomic features. The data application is thorough, with examples from large GWAS (UK Biobank) integrating predicted variant function scores (GenoCanyon) and epigenomic features (H3K27ac marks).

Overall I find the manuscript well-written. The method, albeit a seemingly straightforward extension to a recent work by the same research group, is a timely and highly useful contribution to the current "multi-omics" study theme for many complex trait studies. It could be a good addition to the omics data integration toolbox. However, I think the manuscript can benefit a lot with improved Methods section, to emphasize the novelty and contrast more with the published empirical cFDR method. Specifically,

1. In Introduction I suggest adding some narrative to motivate the method with biological applications. Current manuscript simply mentions in passing that the approach can integration functional genomic data. Even for GWAS data integration, would the new method help to integrate Bayes Factors without having to convert them to z-scores? A list of concrete example along with a few citations of existing approaches to integrate such data with GWAS will help the readers appreciate the contribution of the new method.

2. In Methods, please explicitly summarize key differences between Flexible cFDR and empirical cFDR, and why this is relevant to the proposed application -- it helps to simply listing examples of functional annotation scores that are obviously not a mixture of Gaussian. Also please explain what it is meant by "sparse data" and how exactly it may impact the cFDR approach because I can think of versions of Gaussian mixture that handles sparse data; unless I'm wrong about what sparse data means in this context.

3. One improvement is the use of a flexible threshold rather than a hard threshold of 1/2 when computing cFDR. I can think of a specific version of cFDR without using such a threshold as described in the manuscript, but still uses mixture Gaussian for the auxiliary data. Is that preferable for when the secondary data is also GWAS results, or the new flexible cFDR should replace the empirical cFDR approach for all applications?

4. A concern on line 169 is that it uses a Gaussian kernel. While this is a common KDE choice, is it good enough for arbitrary distribution of secondary data (which can be a positive score, a probability, a binary variable or on phred scale)? Please justify.

Also some additional high level comments:

1. Alternative to cFDR, various multivariate analysis approaches (eg bi-variate GWAS) and data integration methods (eg fGWAS or those based on stratified LDSC) can perform data integration. These are completely different, but possibly more popular alternatives, to cFDR. Please discuss cFDR in the context of these methods.

2. Related to comment above, one limitation of cFDR is that it seems to allow for one auxiliary data-set at a time. Please comment on applications where multiple functional annotations need to be integrated. The application of H3K27ac data integration seems to simply average over different tissues and cell types?

Minor points on the methods:

1. Please briefly recap the definition of boundary effect in KDE.

2. Line 214 please refer to eqn. (9) to clarify how it works.

Reviewer #3: The authors have proposed a new method, flexible cFDR, that generalizes the conditional FDR approach to incorporate auxiliary data information in GWAS of a trait. Usually the auxiliary data incorporated in existing method include p-values from a related trait. Here the authors have leveraged auxiliary data from arbitrary distributions such as functional genomic data. They have done extensive simulation analysis and benchmarking. They also performed two real data analyses with validation to illustrate properties of their approach against existing ones. In my opinion, the authors have done a good job of addressing previous reviewer comments. I have no additional comments; just the 3 very minor comments below.

*Minor Comments*

1. If possible, provide links from which GenoCanyon scores and ChIP-seq counts were downloaded.

2. Suggest that scientific notation be used instead of E-notation for small numbers like p-values.

3. Currently, the article is a tad lengthy. Perhaps the authors can move some details in the Methods and the Results sections to the supplementary.

**Have all data underlying the figures and results presented in the manuscript been provided?**

Reviewer #1: Yes

Reviewer #2: Yes

Reviewer #3: Yes

PLOS authors have the option to publish the peer review history of their article (what does this mean?). If published, this will include your full peer review and any attached files.

Reviewer #1: No

Reviewer #2: No

Reviewer #3: No

---

## [Decision Letter · Decision Letter 1]

24 Sep 2021

Dear Dr Hutchinson,

Thank you very much for submitting your Research Article entitled 'Leveraging auxiliary data from arbitrary distributions to boost GWAS discovery with Flexible cFDR' to PLOS Genetics.

Both reviewers are satisfied with your submitted revision, except for one minor comment from Reviewer 2. Please submit a revised version addressing the issue of any effect of the order of incorporating multiple auxiliary data. While we expect only a minor revision to the manuscript, we require some form of evidence of no effect or some measure of  size and implications of any effect.

We hope to receive your revised manuscript within the next 15 days. If you anticipate any delay in its return, we would ask you to let us know the expected resubmission date by email to plosgenetics@plos.org.

[LINK]

Yours sincerely,

Jingjing Yang, Ph.D.

Guest Editor

PLOS Genetics

David Balding

Section Editor: Methods

PLOS Genetics

Reviewer's Responses to Questions

**Comments to the Authors:**

Reviewer #2: The authors have satisfactorily addressed and clarified my previous concerns. My only remaining question on the method relates to the iterative approach to incorporate multiple auxiliary data. Different from a formal multivariate analysis, the order of data-sets entering the iterative approach seem to matter. If that is the case, is the result sensitive to the order of input? I suggest the authors show with theoretical or empirical results that this is not a concern, or acknowledge it as a short-coming for multiple auxiliary data.

Reviewer #3: I had only minor comments in the previous review, which the authors have addressed. I have no more comment to add.

**Have all data underlying the figures and results presented in the manuscript been provided?**

Reviewer #2: Yes

Reviewer #3: Yes

PLOS authors have the option to publish the peer review history of their article (what does this mean?). If published, this will include your full peer review and any attached files.

Reviewer #2: No

Reviewer #3: No

---

## [Editor Report · Decision Letter 2]

30 Sep 2021

Dear Dr Hutchinson,

We are pleased to inform you that your manuscript entitled "Leveraging auxiliary data from arbitrary distributions to boost GWAS discovery with Flexible cFDR" has been editorially accepted for publication in PLOS Genetics. Congratulations!

Yours sincerely,

Jingjing Yang, Ph.D.

Guest Editor

PLOS Genetics

David Balding

Section Editor: Methods

PLOS Genetics

**Data Deposition**

http://datadryad.org/submit?journalID=pgenetics&manu=PGENETICS-D-21-00900R2

**Press Queries**

---

## [Editor Report · Acceptance letter]

15 Oct 2021

PGENETICS-D-21-00900R2 

Leveraging auxiliary data from arbitrary distributions to boost GWAS discovery with Flexible cFDR 

Dear Dr Hutchinson, 

We are pleased to inform you that your manuscript entitled "Leveraging auxiliary data from arbitrary distributions to boost GWAS discovery with Flexible cFDR" has been formally accepted for publication in PLOS Genetics! Your manuscript is now with our production department and you will be notified of the publication date in due course.

With kind regards,

Anita Estes

PLOS Genetics

On behalf of:
